# Assembly mechanisms of the bacterial cytoskeletal protein FilP

Ala Javadi[1] , Niklas Söderholm[1] , Annelie Olofsson[1], Klas Flärdh[2], Linda Sandblad[1]

Despite low-sequence homology, the intermediate filament (IF)–like protein FilP from *Streptomyces coelicolor* displays structural and biochemical similarities to the metazoan nuclear IF lamin. FilP, like IF proteins, is composed of central coiled-coil domains interrupted by short linkers and flanked by head and tail domains. FilP polymerizes into repetitive filament bundles with paracrystalline properties. However, the cations $Na^+$ and $K^+$ are found to induce the formation of a FilP hexagonal meshwork with the same 60-nm repetitive unit as the filaments. Studies of polymerization kinetics, in combination with EM techniques, enabled visualization of the basic building block—a transiently soluble rod-shaped FilP molecule—and its assembly into protofilaments and filament bundles. Cryoelectron tomography provided a 3D view of the FilP bundle structure and an original assembly model of an IF-like protein of prokaryotic origin, thereby enabling a comparison with the assembly of metazoan IF.

## Introduction

The eukaryotic cytoskeleton is composed of networks of actin filaments, microtubules, and intermediate filaments (IFs), which work together to maintain the shape, motion, and mechanical properties of cells (Fletcher & Mullins, 2010). IFs are the least investigated and most diverse among the major cytoskeletal protein complexes. IF proteins can be categorized in six subclasses based on their primary sequences: type I (acidic keratins), type II (neutral or basic keratins), type III (desmin, vimentin, GFAP), type IV (neurofilaments; NF-L, NF-M, and NF-H), type V (lamin A, B, and C), and type VI (nestin) (Stromer et al, 1987; Lendahl et al, 1990). All IFs share two major biochemical characteristics: first, their capacity to self-assemble in physiological buffers in the absence of cofactors (Godsel et al, 2008; Kelemen, 2017); second, the IF tripartite structural organization, comprising a conserved α-helical rod domain flanked by non–α-helical head and tail domains of different size, sequence, and function (Herrmann & Aebi, 2016; Kelemen, 2017). The α-helical domain of IF proteins form coiled-coil units that polymerize into non-polar filaments (Herrmann & Aebi, 2004). IF polymerization is a hierarchical process based on a combination of lateral and longitudinal associations of the primary units (Herrmann & Aebi, 2004; Block et al, 2015). Filamentation of vimentin and desmin is established in three steps: the first step initiates with rapid lateral association of eight tetrameric subunits forming 60–65 nm unit length filaments (ULFs) (Herrmann et al, 1999); in the second step, filaments grow longer by end-to-end attachment of ULFs (Portet et al, 2009); and the third step consists of compaction, resulting in ~10 nm thick filaments (Stromer et al, 1987). The nuclear IF proteins, lamins, also follow a hierarchical polymerization process, but its filament assembly differs from cytosolic IFs. Head-to-tail attachment of dimers results in thin protofilaments with a "beaded" appearance because of the immunoglobulin-like tail domain (Ben-Harush et al, 2009). Subsequent lateral association of protofilaments leads to thick bundles and a characteristic paracrystal with a 25 nm striation pattern (Stuurman et al, 1998).

With the discovery of the first bacterial IF-like protein, crescentin, essential for the crescent shape of *Caulobacter crescentus*, it became evident that bacteria, like metazoan cells, benefit from these types of cytoskeleton proteins (Ausmees et al, 2003). Despite low-sequence conservation with IF proteins, crescentin has the same tripartite domain organization with a central coiled-coil protein fold (Ausmees et al, 2003). Furthermore, crescentin self-assembles in vitro into long filaments with a diameter of ~10 nm without the presence of cofactors, and, like eukaryotic IFs, the filamentation and solubility of crescentin are pH- and salt dependent (Herrmann & Aebi, 2004; Cabeen et al, 2011).

IF-like proteins from other bacterial species have also been reported, such as CfpA (Izard, 2006), Scc (Mazouni et al, 2006), Scy (Walshaw et al, 2010), and FilP (Bagchi et al, 2008). FilP is an IF-like protein from *Streptomyces coelicolor* (Bagchi et al, 2008; Fuchino et al, 2013; Söderholm et al, 2018). *Streptomyces* are Gram-positive bacteria with characteristics such as polar growth, multicellularity, differentiation, and sporulation (Flärdh & Buttner, 2009; Jakimowicz & van Wezel, 2012; McCormick & Flärdh, 2012). The polar growth of *Streptomyces* hyphae is governed by a protein complex referred to as the polarisome. The coiled-coil protein DivIVA recruits the polarisome to the tip and FilP to its subapical location (Flärdh, 2003; Hempel et al, 2008; Richards et al, 2012; Fuchino et al, 2013; Holmes

[1]Department of Molecular Biology, Umeå University, Umeå, Sweden   [2]Department of Biology, Lund University, Lund, Sweden

Correspondence: linda.sandblad@umu.se

et al, 2013). It has been suggested that FilP provides rigidity and elasticity to the hyphal tip and that deletion of *filP* results in a crooked hyphae phenotype (Bagchi et al, 2008). Like eukaryotic IFs, FilP is composed of the tripartite coiled-coil structure and forms striated filaments in vitro in the absence of cofactors. FilP filaments appear similar to lamin filaments, but the distance between each repetitive unit in the FilP filaments is 60 nm, whereas lamin filaments form a banding pattern measuring ~25 nm (Stuurman et al, 1998; Herrmann & Aebi, 2004; Bagchi et al, 2008). Wild-type crescentin, like vimentin, forms long, nonstriated filaments in vitro. Interestingly, a truncated version of crescentin forms striated filaments similar to FilP and lamin (Cabeen et al, 2011).

In vitro polymerization studies of FilP have earlier been performed on recombinant polyhistidine-tagged fusion proteins. Steric hindrance and charge of fusion protein tags could cause polymerization artifacts or interfere with protein function. For example, FilP–EGFP fusion proteins did not complement a ΔfilP phenotype (Bagchi et al, 2008). In vitro studies of non-tagged FilP have been difficult since removal of the His-tag is not possible because of the insoluble nature of FilP. In this study, we compared N-, C-terminal and non-tagged FilP polymerization in vitro to determine whether a His-tag interferes with polymerization, function or its subcellular localization of FilP in *S. coelicolor*. Unlike the well-characterized metazoan IFs, the basic building units and assembly of IF-like proteins, such as FilP, have not yet been described or visualized. We studied FilP filament assembly in detail using high-resolution imaging techniques in combination with biochemical methods. The combined findings resulted in a model for how FilP coiled-coil building blocks assemble into protofilaments that form paracrystals or gel-like 3D meshworks. The preferred assembly depended on buffer components and pH. Altogether, we propose the first model of filament formation of a prokaryotic IF-like protein.

# Results

### His-tagged FilP expressed in *Streptomyces* localize to hyphal tips, are functional, and assemble into striated filaments

Wild-type FilP localizes close to the tips of growing hyphae in *S. coelicolor*, but because EGFP-fusions interfered this localization (Bagchi et al, 2008; Fuchino et al, 2013), we first addressed whether a fusion with the small polypeptide 6×histidine (6×His) tag affected cellular localization, phenotype, or filament assembly. Recombinant His-tagged and non-tagged FilP were expressed both in *S. coelicolor* and *Escherichia coli*. An immunolabeling protocol with *S. coelicolor* grown on cellophane was developed to examine the localization of N- or C-terminally 6×His-tagged FilP in a ΔfilP-mutant background. Immunofluorescence labeling using α-FilP antibodies revealed a subapical localization for both N- and C-terminally His-tagged FilP, similar to the localization of endogenous FilP in the wild-type strain (Figs 1A–C and S1A–C). The ΔfilP phenotype was characterized with crooked hyphae (Bagchi et al, 2008). Upon induction of His-tagged FilP expression, the N-terminally tagged protein complemented FilP function and displayed straight hyphae. However, the C-terminally tagged protein failed to complement the ΔfilP phenotype (Fig S1D–E).

To visualize the FilP filament structures formed in *S. coelicolor*, native N-terminally 6×His-tagged FilP was affinity-purified from the cytoplasmic fraction of the Δ*filP tipAp-His-N-filP* strain, without disrupting polymers through denaturation. Part of the purified FilP was denatured in 6 M urea and subsequently refolded in assembly buffer (50 mM, Tris–HCl pH 6.8). Both natively purified FilP and refolded FilP were negatively stained with uranyl acetate and imaged by EM to characterize the filaments (Fig 1D–E). The filaments observed in the native fraction and refolded FilP were identical to recombinant FilP expressed in *E. coli*, with the characteristic paracrystalline banding pattern, bridged by transversal protofilaments and similar to lamin paracrystals (Herrmann & Aebi, 2004; Bagchi et al, 2008). The repeating unit of the paracrystal array consists of a thick, protein-dense major band (arrowheads in Fig 1D) and a bridging area containing a less pronounced minor band (asterisks in Fig 1D). The distance of the repeating unit, as measured from the center of one major band to the center of the following major band, for native His-N-FilP bundles was 56 ± 3 nm (n = 330; Fig 1D) and was also 56 ± 3 nm for denatured and refolded FilP bundles (n = 144; Fig 1E). Thus, His-N-FilP expressed in *S. coelicolor* was able to localize to hyphal tips and fulfill the same function as the wild-type protein. Also, protein isolated from *S. coelicolor* formed filaments with the characteristic striated pattern, similar to filaments formed through denaturation and refolding in vitro.

### FilP assembles into filaments independent of protein tags

We purified recombinant N- and C-terminally His-tagged and non-tagged FilP expressed from pET vectors in *E. coli* to investigate the influence of the His-tag on the repetitive striation pattern and paracrystal structure of FilP filaments. To purify non-tagged FilP, a new protein purification protocol was established wherein insoluble FilP was solubilized with lysis buffer containing 6 M urea, and proteins were further separated through ion-exchange chromatography. The purity of His-N-FilP, FilP-C-His, and non-tagged FilP was verified by SDS–PAGE (Fig S2A).

Purified recombinant proteins were dialyzed in assembly buffer in both low and high concentrations (20 and 200 μg/ml), negatively stained, and visualized by EM (Fig 2A–C). His-N-FilP formed filament bundles with a major band periodicity of 60 ± 2 nm (n = 2,141; Figs 2A and S2B), as reported previously (Bagchi et al, 2008). Interestingly, FilP-C-His formed two types of filaments. At the low concentration, filaments were formed with a banding pattern with an average distance of 28 ± 5 nm (n = 1,287) without distinct major and minor bands. Filaments without a striated pattern were formed at the high concentration (Fig 2B). Non-tagged FilP formed thin filaments with a periodicity of 19 ± 3 nm (n = 1,362) at the low concentration (Fig 2C). However, non-tagged FilP formed very thick filament bundles with a major band periodicity of 54 ± 5 nm (n = 694) at the high concentration, similar to that of His-N-FilP (Fig 2C, table of filament banding periodicity in Fig S2C). The additional 18 aa introduced with the 6×His-tag, linker, and protease cleavage sites of the pET28a vector are likely causing the 6 nm longer repeating unit formed by His-N-FilP compared with the non-tagged protein. Upon image alignment of those protein constructs, it became evident that a ~60 nm non-polar repeating unit is a basic feature of FilP filaments (Fig S2C). His-tags associated with the FilP N-terminus resulted in a denser major band, and when associated with the C

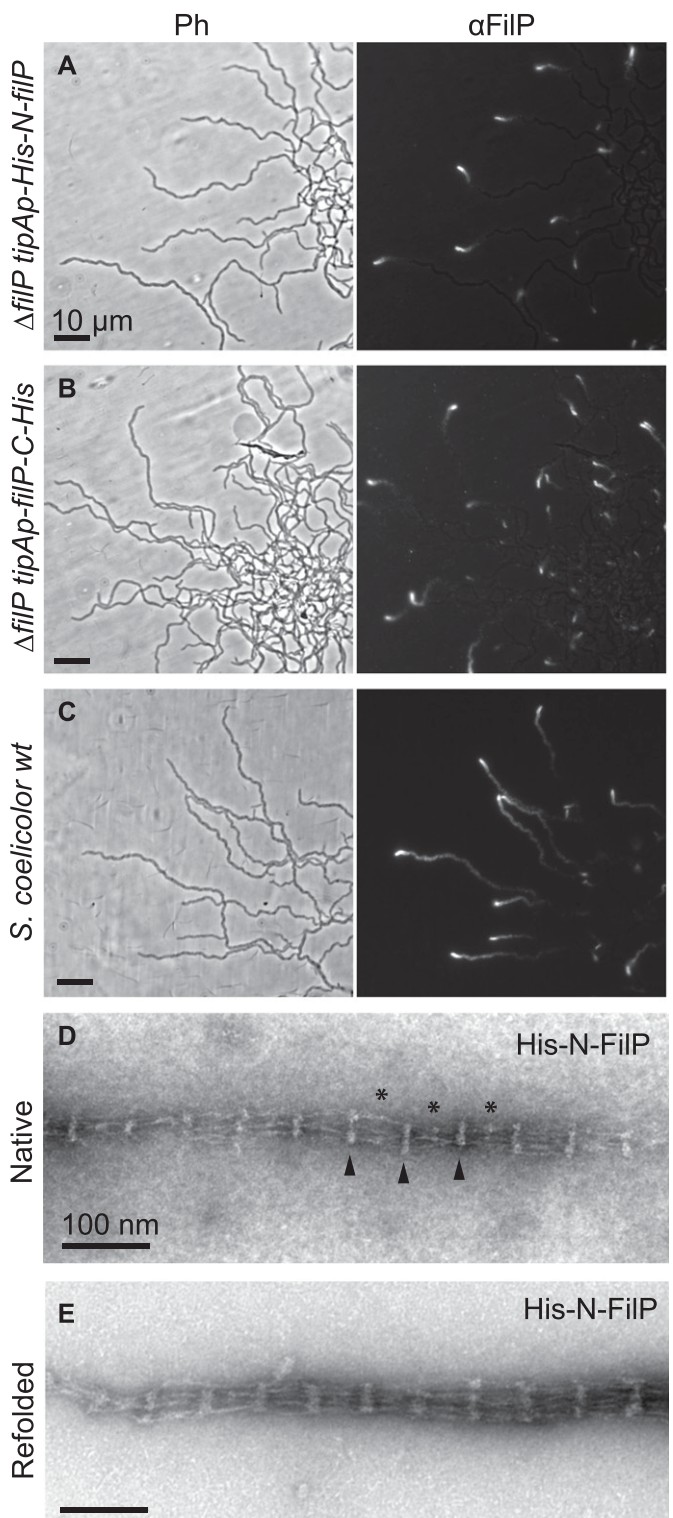

**Figure 1. FilP localizes to the hyphal tips.**
**(A–C)** Immunofluorescence microscopy of hyphae on cellophane reveal tip localization of N-terminally (A) and C-terminally (B) 6×His-tagged FilP in Δ*filp S. coelicolor* background and FilP in wild-type (wt) *S. coelicolor* (M145) strain (C). **(D)** Negative staining EM image of affinity-purified His-N-FilP from strain *S. coelicolor* LS101. The protein was purified and adsorbed to EM grids without denaturation in urea. FilP filament bundle major bands are indicated with black arrowheads and

terminus of FilP, the minor band became denser, giving the filaments two repeats within the 60 mn unit. By removal of the N-terminal head and first coiled-coil domain, a FilP construct comprising aa 71–310 displayed a reduced density in the major band.

N-terminally His-tagged FilP expressed in *E. coli* were chosen for further experiments because of the stable and reproducible structures at low and high concentrations, higher protein yield, and structural similarities to FilP filaments isolated from *S. coelicolor*.

## FilP filament assembly is concentration dependent

EM visualization of His-N-FilP overexpressed in *E. coli* showed a massive paracrystalline structure with the characteristic 60 nm striation pattern in the cytoplasm, demonstrating the potential of FilP to form structures in a cytoplasmic environment (Fig 2D). However, FilP-C-His, with the shorter repeat, and non-tagged protein were not visible as striated structures in the densely stained volumes of *E. coli* (Fig S2D and E). In vitro assembly of the purified FilP protein formed laterally associated structures ranging from single protofilaments up to 1 μm thick filament bundles. Thick paracrystals have, to date, not been observed at the tip of hyphae and, thus, might not be biologically relevant. Therefore, the concentration dependencies of FilP filamentation were addressed using a turbidity assay measuring the light scattering of filamentous polymers at 350 nm. His-N-FilP concentrations ranging from 20 to 200 μg/ml showed a concentration-dependent filamentation (Fig 2E). The critical concentration for FilP filamentation was calculated to be 33 μg/ml. Next, FilP of concentrations ranging from 5 to 200 μg/ml was dialyzed in Tris assembly buffer, negatively stained, and imaged by EM to evaluate the effect of protein concentration on filament thickness and structure. FilP proteins mainly formed thick and straight paracrystals at 200 μg/ml (Figs 2F and S2B for a scanning electron microscopy [SEM] surface view of the paracrystal). However, reduction of the concentration to <20 μg/ml resulted in more branched, bending, and thinner filaments. Occasional filaments could be detected by negative staining EM in samples at 5 μg/ml FilP, a concentration lower than the calculated critical concentration (Figs 2F and S2F).

The cellular FilP concentration in *S. coelicolor* was estimated to relate the in vitro experiment concentrations to the biological concentration. Known concentrations of recombinant purified FilP were used as standards for comparison with whole *S. coelicolor* lysates by Western blotting (Fig S2G). FilP band intensity of the *S. coelicolor* samples corresponded to 57 μg FilP protein/g wet weight mycelia. With an assumption that the wet bacteria density is 1.1 g/ml (Milo, 2013), the cellular concentration of FilP can be estimated to 63 μg/ml. Thus, the endogenous FilP concentration in *S. coelicolor* is within the range of spontaneous filament formation observed in vitro.

## Filamentation is instant upon renaturation of FilP

IFs assemble through the interaction of their coiled-coils into dimers, tetramers, ULFs, and, last, mature filaments (Aebi et al, 1988).

minor bands with asterisks (*). **(E)** Negative staining EM image of affinity-purified His-N-FilP from *S. coelicolor* LS101, unfolded by dialyzing in 6 M urea buffer and refolded by dialyzing into standard Tris assembly buffer, pH 6.8.

Previous studies have shown that eukaryotic IFs, such as vimentin, can exist in their intermediate polymer state; dimers in the presence of 6 M urea or tetramer complexes in the presence of 4.5 M urea (Mücke et al, 2004).

FilP was purified and stored in 6 M urea in its denatured form and dialyzed in Tris-based assembly buffer with a series of urea concentrations ranging from 0.5 to 3 M to induce filamentation, then negatively stained and imaged by EM (Fig 3A–C). No filaments were observed when FilP was dialyzed in buffers containing 3 M urea. Very few filaments, but with the same characteristic banding pattern observed in earlier experiments, were observed in buffer containing 2 M urea. At 0.5 M urea, filaments were generally thinner and with the same repetitive structures as those in the standard assembly buffer without urea (Fig 3A–C). The experiment proceeded by quantifying FilP filamentation using a turbidity assay of FilP samples diluted into assembly buffers with a series of urea concentrations ranging from 0.08 to 3 M (Fig 3D). The quantification showed an onset of filamentation when the urea concentration was less than 3 M during dialysis, which confirmed the negative staining observations.

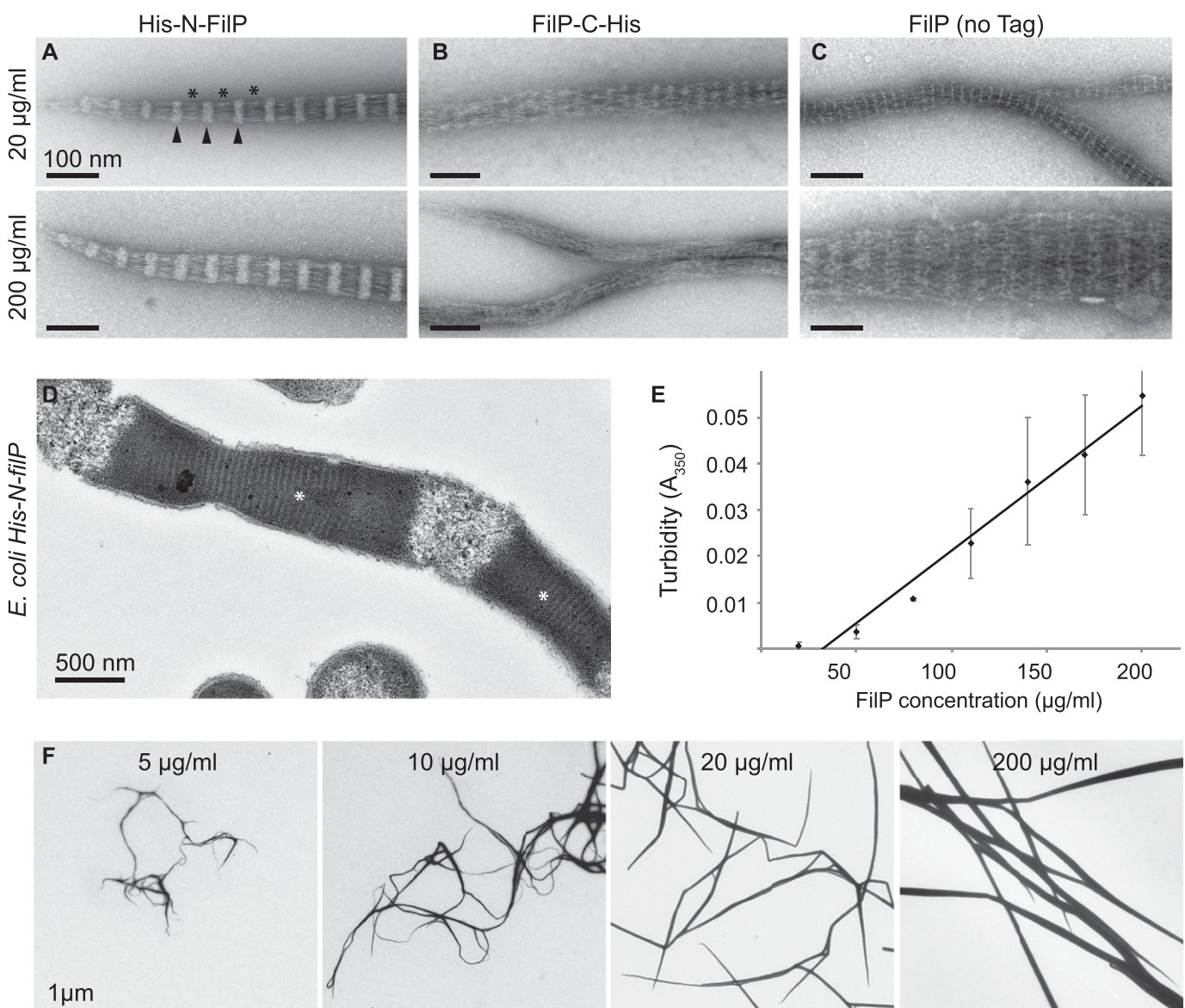

**Figure 2. FilP filament formation is concentration-dependent.**
**(A)** Negative staining EM images of affinity-purified His-N-FilP from *E. coli* (20 and 200 μg/ml) were dialyzed in assembly buffer. Major bands are indicated with black arrowheads and minor bands with asterisks (*). **(B)** Negative staining EM images of affinity-purified FilP-C-His (20 and 200 μg/ml). **(C)** Negative staining EM images of inclusion body washed and ion-exchange purified non-tagged FilP from *E. coli* (20 and 200 μg/ml). **(D)** Thin section EM image of *E. coli* BL21 induced to express His-N-FilP. Cytoplasmic formation of FilP paracrystals is indicated by white asterisks (*). **(E)** Turbidity of His-N-FilP at concentrations of 50–200 μg/ml was measured at 350 nm wavelength after 1 h dialysis against assembly buffer. The average of three experiments is shown, and error bars represent the SD, $P \leq 0.05$, which was calculated using a one-way ANOVA test. **(F)** Low-magnification overview of filament bundles on negative staining EM grids of a His-N-FilP concentration step gradient between 5 and 200 μg/ml displaying straighter filaments and accumulative bundle thickness along with increasing protein concentration.

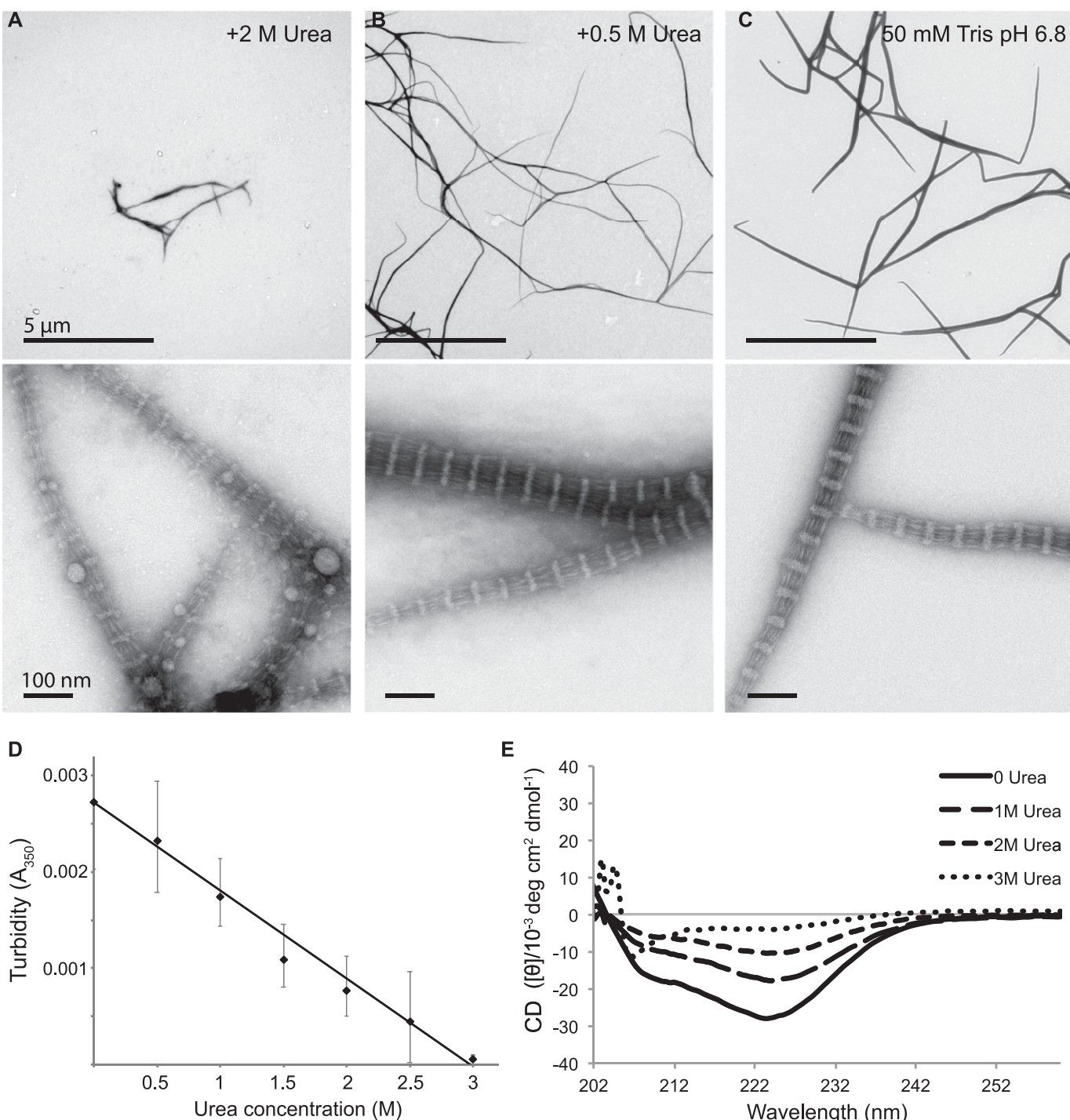

**Figure 3. Filament formation and stability of FilP secondary structure in urea.**
**(A–C)** Negative staining EM images of FilP dialyzed against Tris assembly buffers containing different urea concentrations ranging from 0.5 to 2 M. **(D)** Filament-induced adsorption by turbidity measurement at 350 nm of FilP in increasing urea concentrations shows filament stability in the presence of urea. The average of three experiments is shown, and error bars represent the SD, $P \leq 0.05$, which was calculated using a one-way ANOVA test. **(E)** CD analysis of FilP dialyzed against Tris assembly buffer containing 0–3 M urea shows the stability of the $\alpha$-helical folded structure in the presence of urea.

To test the relation of FilP $\alpha$-helical folding to filamentation, FilP was dialyzed in assembly buffers containing urea ranging from 0.5 to 3 M and analyzed with circular dichroism (CD). CD measurement data showed a gradual increase in FilP $\alpha$-helical structure with reduced urea concentration. A complete absence of secondary structure was observed at 3 M urea (Fig 3E). The maximum urea concentration tolerated for FilP $\alpha$-helical stability correlates with the urea concentration, 3 M, allowing filamentation. Together, the

results of EM imaging, CD, and turbidity assays all demonstrated instant filamentation upon folding when the urea concentration in the sample dropped below 3 M. No basic building blocks could be observed using gradients of urea. In an attempt to capture basic building block units, we chemically fixed the samples at 10 and 30 min of dialysis into phosphate assembly buffer. Thin protofilament-like structures without distinct ULFs or stable, smaller complexes were observed at 10 min (Fig S3A). After 30 min, the protofilaments had grown longer but bundled less than in Tris assembly buffer (Fig S3B). However, at 10 min, initiation of filamentation had already occurred, and further attempts to capture pre-filamentation stages proved unsuccessful, indicating that filamentation of FilP is a rapid process.

## FilP filament assembly is pH dependent

It has been demonstrated that buffer composition has a major impact on the solubility of eukaryotic IFs (Herrmann & Aebi, 2004). Cytoplasmic IFs, such as vimentin and keratin, are soluble in buffers with low ionic strength and high pH (Herrmann & Aebi, 2004). However, nuclear IF lamin is soluble in buffers with high ionic strength and high pH (Aebi et al, 1986; Herrmann et al, 2002). The IF-like protein crescentin showed the same behavior as the cytoplasmic IFs, being more soluble at high pH and lower ionic strength (Cabeen et al, 2011). To clarify the effect of pH and ionic strength on FilP solubility and filament formation, three different buffers and pH (citric acid pH 3.9, Tris pH 8.8, and ethanolamine pH 11) were compared with standard assembly buffer by negative staining EM and an ultracentrifugation sedimentation assay. EM analysis revealed that FilP formed filaments with banding patterns of 60 ± 4 nm (n = 169) at pH 3.9. However, the filament banding pattern appeared vague, and the filament bundles were thin (Fig 4A). Low pH altered the net charge of the FilP, which could affect the lateral alignment of protofilaments and explain the less pronounced banding pattern observed. FilP at pH 8.8 formed fewer and less condensed filaments compared with assembly buffer control at pH 6.8 (Fig 4B and C). No filaments were observed at pH 11; instead, FilP formed rod-shaped polymer units (Fig 4D). The degree of FilP solubility in these samples were addressed by a sedimentation assay. Total sedimentation of FilP was observed at pH 3.9 and 6.8 (Fig 4E). However, FilP solubility increased with rising pH. At pH 8.8 and 11, 32% and 72% of total FilP was soluble, respectively. In summary, both filament assembly and banding pattern were affected by pH, and FilP was more soluble in alkaline buffers, as observed for metazoan lamin and cytosolic IFs.

## FilP forms meshwork in the presence of monovalent cations

Different buffer conditions have drastic effects on in vitro filament formation of IFs. For example, in vitro studies have demonstrated that phosphate buffers can unravel the filamentous structures of nuclear and cytoplasmic IFs such as desmin, neurofilaments, and lamin (Aebi et al, 1988). To study the effect of different buffer systems on FilP filament assembly, the samples were dialyzed in the buffers for 30 min before imaging by negative staining EM. Three buffers at neutral pH were compared with the control assembly buffer (50 mM Tris): 20 mM Hepes, 50 mM sodium phosphate, and a

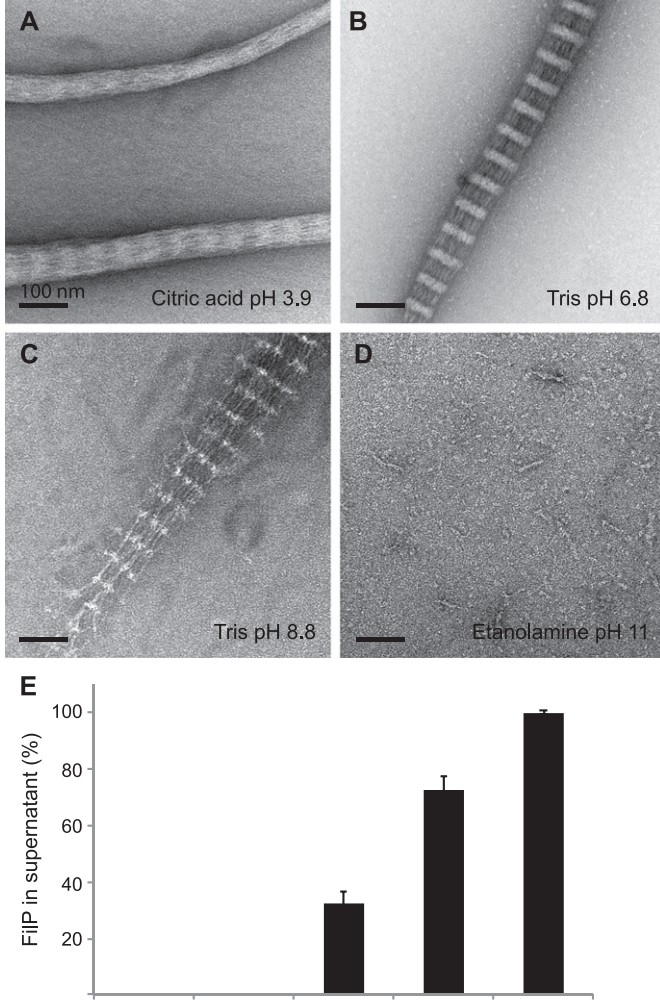

**Figure 4. FilP filament formation is affected by pH.**
**(A)** Negative staining EM images of FilP dialyzed in citric acid, pH 3.9, resulted in thicker filaments with a diffused banding pattern (e.g., the lower filament in this image) and condensed thinner filaments without striations (e.g., the upper filament in this image). **(B)** FilP dialyzed in standard Tris assembly buffer, pH 6.8, formed paracrystals with a banding pattern. **(C)** FilP dialyzed in Tris, pH 8.8, formed less compact striated filaments. **(D)** FilP dialyzed in ethanolamine, pH 11, formed rod-like structures. **(E)** FilP solubility measured by ultracentrifugation after FilP dialysis in buffers with pH ranging from 3.9 to 11. Bars represent protein content in the supernatant. The average of three experiments is shown, and error bars represent the SD, $P \leq 0.05$, which was calculated using a one-way ANOVA test.

cytoplasm-mimicking buffer called polymix (Jelenc & Kurland, 1979). Dialysis resulted in the characteristic banded FilP filaments, but FilP assembled into thinner filaments in 50 mM sodium phosphate and Hepes buffer (Fig S3B and 6B). Prolonged dialysis in the presence of phosphate caused dissociation of filaments into oligomeric structures. FilP dialyzed against the polymix buffer formed a mixture of thin protofilaments and a hexagonal 2D meshwork, indicating that FilP polymers can coexist as both filament bundles and meshwork (Figs 5A and S3C).

The hexagonal configuration of FilP has previously been observed, but the buffer components that promote this assembly were not

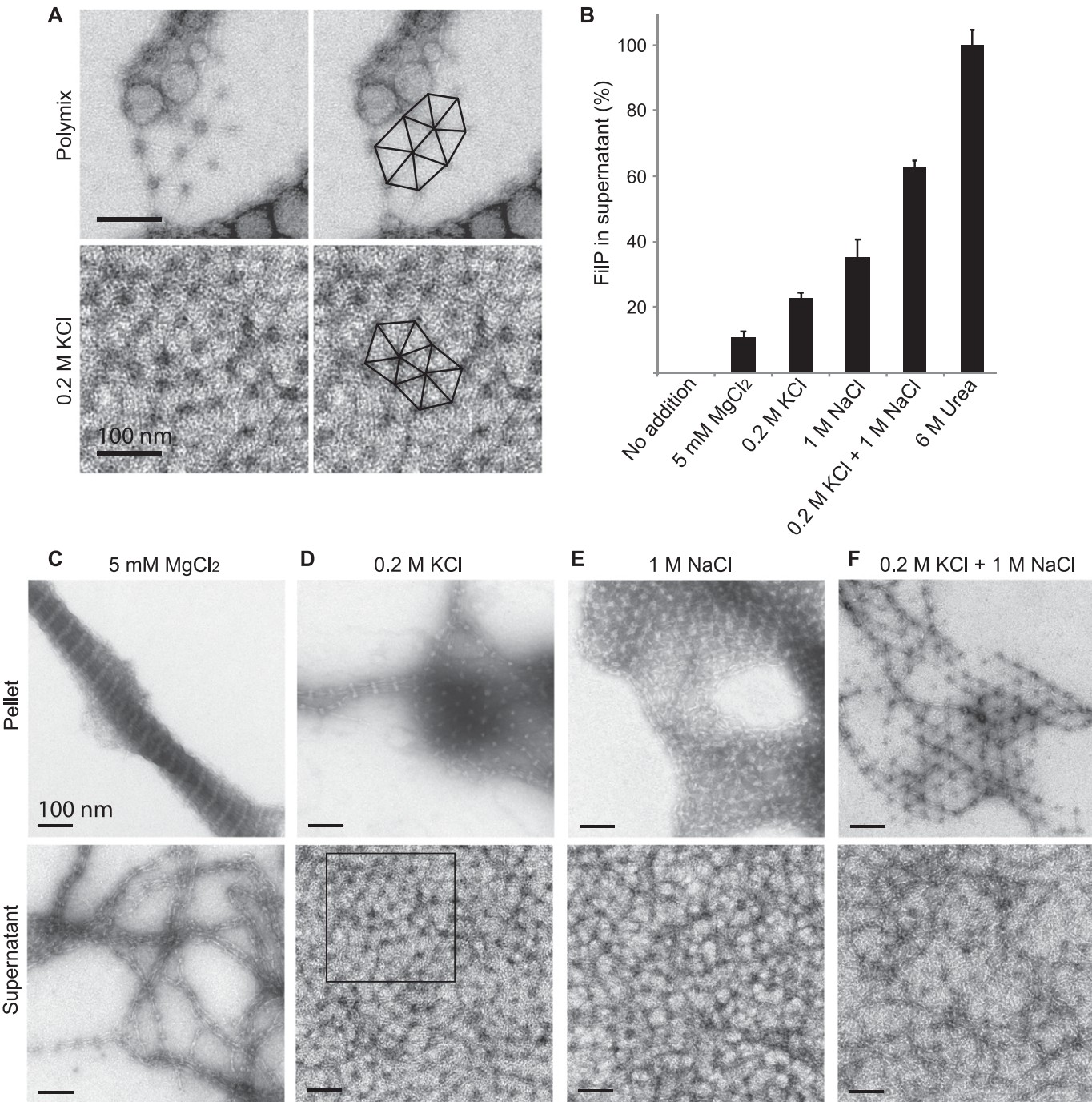

**Figure 5. Cations ($Mg^{2+}$, $Na^+$, and $K^+$) affect FilP filamentation and solubility.**
**(A)** Negative staining EM image with an outlined view of the hexagonally shaped FilP meshwork in polymix buffer (magnified from Fig S3C) and 0.2 M KCl buffer (magnified from D). **(B)** Sedimentation assay; bars indicate the amount of soluble FilP recovered in the supernatant after centrifugation upon dialysis of FilP into assembly buffers without and with additions of cations (5 mM $Mg^{2+}$, 1 M $Na^+$, and/or 0.2 M $K^+$) and also buffer containing urea. The average of three experiments is shown, and error bars represent SD, $P \leq 0.05$, which was calculated using a one-way ANOVA test. **(C)** EM images of pellet and supernatant of FilP dialyzed against assembly buffer containing 5 mM $MgCl_2$. **(D)** EM images of pellet and supernatant of FilP dialyzed against assembly buffer containing 0.2 M KCl. The selected box is magnified in Fig 5A. **(E)** EM images of pellet and supernatant of FilP dialyzed against assembly buffer containing 1 M NaCl. **(F)** Negative staining EM images of pellet and supernatant of FilP dialyzed against assembly buffer containing 0.2 M KCl and 1 M NaCl.

determined (Fuchino et al, 2013). Given the extensive use of $Mg^{2+}$, $K^+$, and $Na^+$ in the in vitro solubility and filament assembly studies of IFs and the IF-like protein crescentin (Herrmann et al, 2002; Foeger et al, 2006; Cabeen et al, 2011), we investigated the effects of these di- and

monovalent cations on the self-assembly and polymer structure of FilP. By an ultracentrifugation sedimentation assay, no FilP could be recovered from the supernatant of a sample in standard Tris assembly buffer. The addition of cations increased FilP solubility, as

observed by both sedimentation assay and EM. Addition of 5 mM $Mg^{2+}$ increased FilP solubility and resulted in 11% recovery of added protein in the supernatant. In the presence of monovalent 0.2 M $K^+$ and 1 M $Na^+$, 25% and 35% of protein could be recovered in the supernatant, respectively (Fig 5B). FilP solubility was significantly enhanced when $Na^+$ and $K^+$ were combined, resulting in a 63% recovery of FilP in the supernatant (Fig 5B).

The insoluble FilP pellet contained thick filament bundles in Tris buffer and the presence of divalent $Mg^{2+}$ (Fig 5C). FilP dialyzed against buffers containing monovalent cations $K^+$, $Na^+$, and the combination of $K^+$ and $Na^+$ formed meshworks comprising connected hexagonal modules, similar to that observed in Polymix (Fig 5D–F). Unlike previous studies of IF and IF-like proteins, we examined both the post-ultracentrifugation pellet and the supernatant by negative staining EM. In earlier works, the supernatant had been assumed to contain soluble and non-filamentous protein (Foeger et al, 2006; Cabeen et al, 2011). In the presence of $Mg^{2+}$, negative staining EM displayed that the supernatant contained thin FilP filament bundles (Fig 5C). FilP dialyzed against buffers containing $K^+$, $Na^+$, and the combination of $K^+$ and $Na^+$ contained meshworks comprising connected hexagonal modules, similar to that observed in polymix (Fig 5D–F). Interestingly, not only the pellet contained polymer structures but also the supernatant placed on a grid exhibited a continuous hexagonal array, which could represent a gel-like conformation of FilP (Fig 5D–F). Using the negative staining method, the proteins are blotted and dehydrated before EM imaging, which causes a flattened appearance in multiple layers of the 3D meshwork. Thus, these results clearly demonstrate the effect of salt in the buffer on FilP filament assembly and structure.

### FilP lateral and longitudinal interactions form filament bundles

The eukaryotic cytoplasmic IFs, such as vimentin and keratin, assemble through end-to-end association of ULFs into higher order filamentous structures (Portet et al, 2009). However, head-to-tail attachment of lamin dimers causes the formation of structures called protofilaments. Lamin protofilaments evolve into thick bundles by lateral association of protofilaments (Aebi et al, 1986; Heitlinger et al, 1991).

To uncover the mechanism of FilP filament formation, we made use of the various polymer conformations identified in this study. First, the rod-shaped structures formed by FilP in ethanolamine buffer, pH 11, were analyzed in greater detail by single-particle analysis of negatively stained samples. Three classes of 316 individual molecules were aligned, and the average length of the three classes was a 37-nm long, rod-shaped molecule (Fig 6A). Interestingly, this result resembles the bioinformatically estimated length of a coiled-coil rod domain predicted using the COILS software (Lupas et al, 1991). COILS predicted that 265 aa of FilP would form coiled-coil domains, corresponding to a theoretical length of 40 nm. Building on these calculations, we suggest that the rod-shaped FilP molecules constitute the primary unit before FilP filament elongation. FilP filaments formed at the critical concentration in Hepes buffer yielded long protofilaments. The protofilaments displayed a repeating "bead on a string" pattern with the 60 nm repeats (Fig 6B). Thus, this suggests that the 60-nm repeating unit could be composed of two rod-shaped molecules

longitudinally interacting tail-to-tail as the non-polar primary stage of assembly.

To verify the localization of N- and C-terminal head and tail position within a protofilament, FilP filament bundles were labeled with nickel-coated 1.8 nm gold beads and imaged by cryo-EM. On His-N-FilP filaments, nano-gold localized specifically to the vicinity of the major bands. However, on FilP-C-His bundles, the localization is less specific, and the addition of nano-gold seams to disassemble the bundles. A faint affinity for nickel-coated gold to FilP filaments was present also on non-tagged FilP bundles (Fig S3D–F). Together, these images indicate that FilP N-terminal heads are gathered in the electron-dense major band and the C-terminal tail-to-tail interaction makes up the minor band (Fig 6B).

### Cryo-electron tomography visualization of FilP filament

Cryo-electron tomography was used to obtain a 3D view and model of the assembled filaments. The reconstructed volume contained branched FilP filament bundles and was visualized by isosurface rendering (Fig 6C). Manually modeled protein density of the 3D volume shows a lateral association of protofilaments aligned repetitively. The electron-dense overlap zones form major bands, and the less dense bridging areas between two major bands constitute the minor bands (Fig 6D and Video 1). Proteins are tightly packed in the major bands (drawn as a green mesh in the model). Within the major bands, no internal symmetrical organization was found, and the resolution does not allow for separation of protofilaments within the major bands in either the X/Y or Z/Y view projection of 20 slices in X of the major band density (II in Fig 6D). In between the major bands, visible and clearly separated protofilaments are drawn as blue tubes in the model. The protofilaments transversally bridge the repetitive elements longitudinally (I). In the center of each repetitive element of the tomogram, protofilaments appeared slightly denser, forming the minor band, which was projected of 20 slices in X in the minor band area (I in Fig 6D). When the protofilaments were modeled as blue rods, FilP subunits were recognized to partially overlap in the minor band. However, the longitudinally bridging protofilaments were not straight according to the bundle axis but, rather, tilted and crossed over each other in the bundle. This cryo-electron tomogram demonstrated how protofilaments interact laterally to form higher order filament bundles in the later stages of assembly.

## Discussion

The *S. coelicolor* IF-like protein FilP forms diverse polymer structures and shares biochemical characteristics with other IFs and IF-like proteins. The striated filaments and paracrystals formed by self-assembled FilP are similar to those reported for lamin, rootletin, and mutated crescentin (Heitlinger et al, 1991; Yang et al, 2002; Cabeen et al, 2011). However, molecular assembly, especially longitudinal interaction, seems to differ among the filamentous proteins with similar filament bundle structure and paracrystal formation ability. We show that FilP also can form a meshwork in physiologically mimicking polymix buffer (Figs 5A and S3C) and in

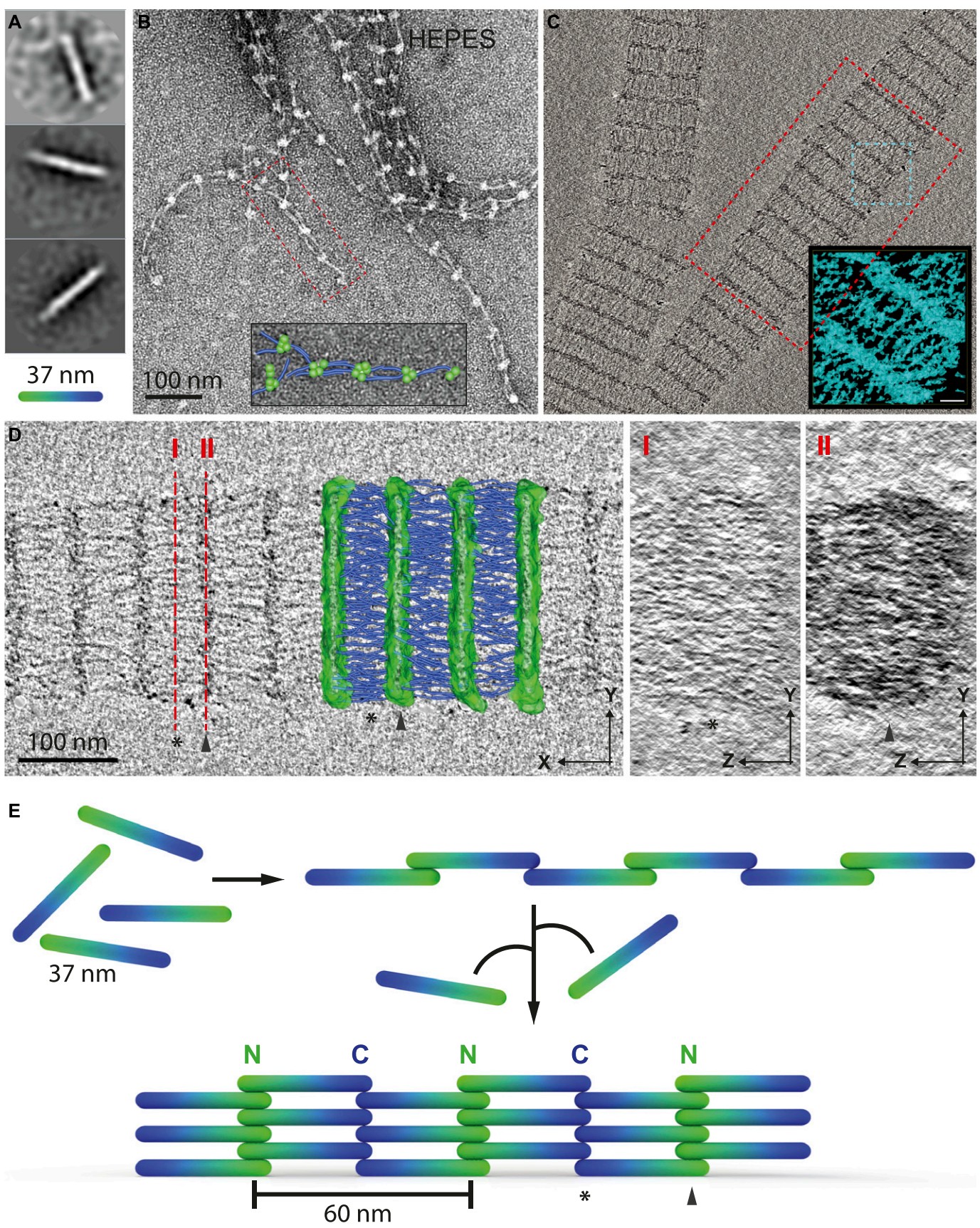

the presence of monovalent K⁺ and/or Na⁺. EM studies of cellular lamin configuration display a meshwork supporting the nuclear membrane, in the form of a woven network of protofilaments or loosely connected shorter oligomers (Aebi et al, 1986; Harapin et al, 2015; Mahamid et al, 2016; Turgay et al, 2017). However, unlike lamin, the FilP meshwork is composed of interconnected hexagonal protofilaments with the same repetitive 60 nm unit length as the banding pattern of filaments (Fig 5A). Our combination of biochemical analyses and EM techniques allowed for a systematic investigation of the parameters that affected in vitro FilP assembly and resulted in a model where this IF-like protein forms filaments and meshworks. The model is outlined in Fig 6E and further discussed in the following paragraphs.

The hydrophobic coiled-coil structure of FilP promotes filamentation as an instant effect of the α-helical fold. As seen during dialysis and refolding experiments from urea to more physiological conditions, FilP polymers are transient at their intermediate steps. The buffer ethanolamine (pH 11) was the only buffer condition without denaturing agents that maintained FilP in a nonfilamentous form in the supernatant after ultracentrifugation of the samples. Under such alkaline conditions, negative staining EM of the soluble FilP exposed rod-shaped particles. Single-particle classification revealed that each rod-shaped particle was 37 nm long—a length closely resembling the predicted theoretical 40 nm of FilP coiled-coil domains. We speculate that each rod-shaped molecule could be a basic building unit formed of a polar and parallel coiled-coil dimer (Fig 6E). Polar dimer structures have also been observed in metazoan IFs as their primary filament-forming unit (Herrmann & Aebi, 1998). Because the distance between repeating units in FilP polymers and hexagonal meshwork is nearly 60 nm, each repeating unit in the filaments might consist of two 37 nm long, rod-shaped molecules, associated tail-to-tail, partly longitudinally staggered during assembly.

Both endogenously and recombinantly expressed His-tagged and non-tagged FilP efficiently formed filaments in vitro. Nevertheless, the protein-dense major and minor bands, as visualized by negative staining EM, differed when filaments were formed from differently tagged FilP. The N-terminally tagged FilP showed a distinct striation pattern similar to non-tagged FilP assembled at high concentrations. N-terminally tagged FilP displayed filament bundles with a 60 nm repeating pattern independent of concentration, suggesting that positively charged amino acids of the tag have a stabilizing effect on the paracrystal structure. The more pronounced major band in the filaments, formed by N-terminally tagged FilP, indicates that the N-terminal head domains are contributing to the additional density of the major bands and that the head domains of FilP are localized in the major bands. Non-tagged FilP at high concentrations and FilP lacking the first N-terminal coiled-coil showed the same repetitive length in the striation pattern, but with significantly lower protein density in the major band, when compared with N-His-FilP bundles (Fig S2C), supporting the model of assembly with FilP N-termini contributing to the major bands. Nickel-coated nano-gold should specifically label the submolecular position of 6×His-tags; this affinity experiment gives a clear picture of the localization of N-terminus of FilP to the major band. Shorter repeating units of 28 nm in the filaments formed by C-terminally tagged FilP, as well as 19 nm in low concentrations of non-tagged FilP filament, are repetitive units with approximately half the length compared with N-tagged FilP. This could be explained if the C-terminus 6×His-tag fusion caused a higher density in the minor band. Consequently, the C-terminal tail of FilP is thought to be positioned in the minor band. By cryo-EM, nickel-coated nano-gold were detected to the striation bands of FilP-C-His bundles, but the intramolecular localization was more scattered and nano-gold displays a faint affinity also to bundles of non-tagged FilP, both to the bands and to the less protein-dense volume with transversal protofilaments. Presumably nano-gold has a background affinity to FilP protein independent of the nickel-6×His interaction, which makes the C-terminal localization unspecific. The existence of filaments with little or no striation pattern in C-terminally tagged FilP indicates the importance of the FilP C-terminal tail for lateral interaction of protofilaments. In summary, the different structural effects of N- and C-terminal His-tags are the basis for suggesting how the primary assembly units, constituted by the FilP dimers, are oriented in the polymer FilP filament bundle structures in our model (Fig 6E).

The rod-shaped primary FilP units, further associate longitudinally in a tail-to-tail and head-to-head manner to form nonpolar, with a 60 nm repeat, beaded protofilaments (Fig 6B). The non-polar FilP protofilament assembly is likely to be more similar to vimentin and desmin longitudinal assembly of their ULFs, rather than to the polar head-to-tail assembly of lamin protofilaments (Herrmann & Aebi, 2004; Ben-Harush et al, 2009). Based on the negative staining EM images of FilP protofilaments, we created a model with FilP N-terminal region (drawn as green density) and the coiled-coil domain and C terminus of the proteins (drawn as blue rods; shown as an insert in Fig 6B). During continuous dialysis at high protein concentration, FilP filament assembly continues via lateral

**Figure 6.  Intermediate stages of FilP filament assembly and a model of bundle formation.**
**(A)** Single-particle 2D class averages from negative staining images of FilP dialyzed in 20 mM ethanolamine, pH 11, buffer, which shows 37 nm rods representing the primary state of filament assembly. Particle images are cropped and rotated for 2D classification, the class averages are consequently displayed as circular images on an average grey background. **(B)** Negative staining EM of FilP dialyzed in Hepes buffer display FilP in the form of individual protofilaments and thin bundles. "Bead on a string" structures are modeled as a head-to-head and tail-to-tail association of FilP rods, resembling protofilaments in the selected magnified box. Green is used to highlight the densities of the N-terminal His-tag, and blue color is used for the rod coiled-coil, including the C-terminal portion of the protein. **(C)** Tomogram central slice of FilP filament bundles/paracrystals show protofilaments in longitudinal, branching, and lateral association. In the selected turquoise box, a subvolume is visualized by IMOD isosurface rendering (scale bar = 20 nm). **(D)** Tomogram selected red box from (C) showing both a projection of 20 pixels in Z from the center of the volume and a model, drawn in 3Dmod, of tomogram protein densities. Green mesh represents the densities of the protein-dense major bands (containing the N-terminus of FilP) and blue rods represent the protofilament transversally bridging between bands. Along I and II, the tomogram volume was rotated 90° around Y to show a projection in X of the major band (I) and the minor band (II) volumes, lacking apparent structural pattern or crystal packing. **(E)** Model of FilP assembly: 37 nm rod domains proposed to consist of FilP parallel coiled-coil dimers assembled into protofilaments. Lateral association of additional primary assembly units results in thick filament bundles and protofilaments with a 60 nm repetitive striation pattern.

association of protofilaments, resulting in varying thickness of the filament bundles. Thick and dense paracrystals were observed to a higher degree in high protein concentrations, in assembly buffers with low ionic strength, in the presence of $Mg^{2+}$, and the absence of phosphate. Moreover, cryo-electron tomography was used to visualize the 3D architecture of the assembled FilP filament bundles. The 3D volume shows that individual protofilaments are clearly recognized, resembling the negative staining EM results. This is an improvement in resolution compared with tomograms of IFs assembling into similar paracrystals in vitro (Ben-Harush et al, 2009). The paracrystal is repeatedly packed along the longitudinal axis, but no symmetry could be detected in the lateral association between protofilaments—in the minor (I) or major (II) bands of the filament bundle (Fig 6D). Instead, when filament densities were individually modeled in the 3D volume, it was evident that no protofilament possesses the same angle as the longitudinal axis of the bundle. Rather it appeared that all protofilaments bridge from one major band to the next with a tilted angle, giving the impression that protofilaments cross over each other in the model projection. The FilP filament bundle seems similar to the lamin paracrystal model; however, lamin was predicted to have more symmetrical packing, with laterally associated protofilaments in line with the bundle axis (Ben-Harush et al, 2009). The 3D arrangements of FilP in a paracrystal-like bundle describes how the filaments assemble; protofilaments are longitudinally very accurately repetitive, but the lateral associations are more flexible, allowing bending and branching within bundles of varying thickness.

Structural variations, bundle thickness, and filament straightness of the FilP filament were observed to be protein concentration dependent. Self-assembled FilP formed paracrystals both in vitro and in the cytoplasm of *E. coli*. To understand whether the concentrations used in vitro were of biological relevance, the concentration of FilP in *S. coelicolor* cytoplasm was estimated to be 63 μg/ml, which is within the range of spontaneous self-assembly observed in vitro, making it reasonable to expect similar repetitive filamentous structures in the bacteria. Earlier studies of FilP have shown specific subapical localization of FilP in the growing hyphae (Fuchino et al, 2013). Here, the measured in vivo concentration comprises an average cytoplasmic concentration of FilP. Therefore, local accumulation of FilP in the subapical area of the tip might be of a higher concentration, thereby feasibly promoting the formation of FilP filament bundles or meshwork.

FilP, like lamin, is more soluble in alkaline buffering conditions and the presence of monovalent cations, such as $K^+$ and $Na^+$. However, FilP filament bundles are more resistant towards $Na^+$ compared with lamin, which was fully solubilized at 300 mM $Na^+$ (Foeger et al, 2006), whereas FilP was only soluble to 40% in 1 M $Na^+$. In addition to the FilP sedimentation assay, we performed negative staining of both supernatants and pellets from the various buffering conditions. We observed higher order structures in the supernatants from the sedimentation assay in the presence of monovalent cations, which was considered to be soluble monomeric protein in previous studies on other IF and IF-like proteins (Foeger et al, 2006; Cabeen et al, 2011). The FilP structures in the supernatant were visualized as a 3D hexagonal meshwork with a gel-like attribute, which could explain its lack of sedimentation during ultracentrifugation. Unlike $K^+$ and $Na^+$, which promoted the formation of a hexagonal FilP meshwork, in the

presence of divalent cation $Mg^{2+}$, the supernatant contained long and thin protofilaments. The $Mg^{2+}$ pellet consisted of striated filaments similar to that of the Tris assembly buffer alone. $Mg^{2+}$-induced filament formation is shared among metazoan IFs, such as lamin and keratin (Foeger et al, 2006; Kayser et al, 2013) and the IF-like protein crescentin (Cabeen et al, 2011). For vimentin, $Mg^{2+}$ increased lateral interaction of protofilaments and reduced longitudinal extension (Herrmann et al, 1999; Cabeen et al, 2011; Brennich et al, 2014). Based on our in vitro EM analysis of the mesh-like structure formed by FilP, we propose that either a 2D hexagonal meshwork along a membrane or a 3D meshwork filling the cytoplasmic volume could be formed at the hyphal tips. Further high-resolution imaging studies of the growing hyphae are required to determine the actual nature of these protein structures.

This study provides a deeper understanding of how FilP assembles into filaments. IFs and IF-like proteins possess certain distinct similarities, such as the tripartite structural architecture and biochemical properties. For instance, aside from the similarities of coiled-coil organization with lamin and crescentin, we showed that FilP also shares considerable solubility and filament assembly properties with both crescentin and lamin (Ausmees et al, 2003; Herrmann & Aebi, 2004; Cabeen et al, 2011). Sterical hindrance, such as stiffness and large dimensions, makes paracrystals structures less biological relevant, even if large aggregates were formed and observed in *E. coli*. The molecular interactions towards FilP in the bacteria, for example, binding proteins, association to membrane or carbohydrates are aspects, which may have impact on the tertiary structure. A more complete view of the spatial organization of the bacterial IF-like cytoskeleton requires further cellular investigations. However, in this in vitro protein model, FilP shows sufficient flexibility to assemble into both bundles and more biologically adaptable 2D or 3D meshworks.

Several bacterial species have been shown to contain IF-like proteins, such as CfpA, AglZ, and Scc (Yang et al, 2004; Izard, 2006; Mazouni et al, 2006). The diversity and irregular distribution of the IF-like proteins among various prokaryotic genomes suggests an independent origin of IF-like proteins for each lineage and convergent evolution of orphan genes (Preisner et al, 2018). Further combined functional and structural studies on these diverse IF-like proteins are required to explore whether these conformational traits are specific to FilP or common to bacterial IF-like proteins.

# Materials and Methods

### Cloning

The bacterial strains, plasmids and primers used in this study are listed in Table S1. Cloning in this work has been performed according to standard protocols, and constructs were verified by nucleic acid sequencing. Genes for non-tagged FilP, N-terminal FilP, and C-terminal FilP were cloned in pETM13, pET28a, and pET21a, respectively. *E. coli* strains were grown in lysogeny broth or on lysogeny agar plates containing ampicillin (100 μg/ml) or kanamycin (50 μg/ml). *S. coelicolor* strains were grown on tryptone soy broth (TSB) or tryptone soy agar (TSA) plates. N- and C-terminally

tagged FilP were introduced into *S. coelicolor* by conjugation into plasmid pIJ6902 as described in Kieser et al (2000).

### Light microscopy

Phase-contrast imaging of *Streptomyces* hyphae was generally performed as described by Kieser et al (2000). *S. coelicolor* spores were cultured on TSA plates containing apramycin (50 $\mu$g/ml), and addition of thiostrepton (10 $\mu$g/ml) for expression from the pIJ6902 plasmid. For immunofluorescence, the spores were inoculated on cellophane placed on TSA plates and incubated for 16 h at 30°C. The cellophane was removed from the TSA and placed on a hydrophobic plastic surface. Fixing solution (2.8% PFA, 0.0045% GlutA in PBS) was quickly applied on the cellophane and incubated for 15 min at RT. Next, the cellophane was washed with PBS buffer before covering with permeabilization buffer (20 mM Tris–HCl, pH 8, 50 mM glucose, 10 mM EDTA, and 2 mg/ml lysozyme) for 1 min. After washing the cellophane with PBS buffer, it was covered with blocking solution (1× PBS, 2% BSA) for 5 min, washed with PBS, and incubated with primary polyclonal rabbit anti-FilP antibody (Söderholm et al, 2018) diluted 1:1,000 in blocking buffer. Afterwards, the cellophane was washed with PBS and incubated with goat anti-rabbit, Alexa Fluor 647 (Life technologies) secondary antibody diluted 1:1,000 in blocking buffer for 1 h. The cellophane samples were washed with PBS and mounted onto agar pads for imaging. Complementation studies of the Δ*filP* phenotype were performed by inoculating spores in the acute angle of glass coverslips inserted into TSA plates containing appropriate antibiotics and incubated for 48 h at 30°C before analyses. Hyphae protruding upwards onto the coverslip were used for phenotypic analysis, a culturing method adopted from Bagchi et al (2008). Imaging was performed with Nikon Eclipse 90i fluorescence microscope equipped with appropriate filter sets, a Hamamatsu ORCA-ER camera, and NIS Elements AR software. Images were cropped and contrast adjusted using Adobe Photoshop CC software.

### Protein purification

#### *Affinity purification from* S. coelicolor *culture*

*S. coelicolor* strain Δ*filP tipAp-His-N-filP* spores were inoculated in TSB containing 5 $\mu$g/ml thiostrepton and incubated at 30°C for 48 h. Bacteria were harvested through centrifugation, flash-frozen in liquid nitrogen, and thawed on ice, resuspended with lysis buffer (0.01 M sodium phosphate buffer, 0.01 M Tris–HCl, pH 6.8, protease inhibitor cocktail tablets [Roche], and 0.5% Triton X-100 [Sigma-Aldrich]), incubated for 1 h at RT, and lysed through sonication. The lysate was cleared through centrifugation at 10,000$g$ for 30 min. Protein-containing supernatant was incubated with TALON metal affinity resin (BD Biosciences) for 30 min at RT and washed with (0.01 M sodium phosphate buffer, 0.01 M Tris–HCl, pH 6.8, and 5 mM Imidazole [Sigma-Aldrich]). FilP was eluted from TALON resin with elusion buffer (0.01 M sodium phosphate buffer, 0.01 M Tris–HCl, pH 6.8, and 150 mM Imidazole).

#### *Purification of non-tagged FilP*

Overnight culture of *E. coli* strains containing non-tagged FilP in pETM13 plasmids were diluted in 1 liter Lysogeny Broth containing 50 $\mu$g/ml kanamycin to an $OD_{500}$ ~ 0.05. Cultures were incubated at 37°C until they reached an $OD_{500}$ ~ 0.5. Next, IPTG was added to the cultures to a final concentration of 1 $\mu$g/ml, and the cultures were incubated for 3 h in 37°C to induce protein expression. The cultures were harvested and washed twice with ice-cold PBS, flash-frozen in liquid nitrogen, and stored at –80°C. The cells were thawed on ice, resuspended in lysis buffer (50 mM Tris–HCl, pH 6.8, protease inhibitor cocktail tablets [Roche], and 0.5% Triton X-100 [Sigma-Aldrich], 0.1 mg/ml lysozyme from hen egg white [Fluka] and 200 mM NaCl), and lysed with a French press (1.2 kbar). The lysate was centrifuged at 30,000$g$ for 30 min at 4°C. Insoluble FilP in inclusion bodies was sedimented during centrifugation. The supernatant was discarded, and the inclusion bodies were resuspended in lysis buffer containing 6 M urea (Sigma-Aldrich). All of the buffers containing urea were prepared at the time of use. The resuspended sample was then centrifuged at 30,000$g$ for 30 min at 4°C. The supernatant was collected and dialyzed overnight in a 10,000 MWCO Slide-A-Lyzer dialysis cassette (Thermo Fisher Scientific) into binding buffer A (20 mM citric acid, pH 3, and 6 M urea). Samples dialyzed overnight were collected and incubated with SP ion-exchange sepharose (GE Healthcare) at RT for 1 h. FilP was eluted with binding buffer A containing 400 mM NaCl. Eluted FilP was dialyzed overnight against binding buffer B (50 mM Bicine [ChemCruz], pH 8.7, and 6 M urea). Dialyzed FilP was incubated with SP ion-exchange sepharose at RT for 1 h. Pure FilP was collected from the flowthrough. The protein concentration in this experiment and all the other experiments were measured with the Pierce BCA protein assay kit (Thermo Fisher Scientific) and NanoDrop 2000 C (Thermo Fisher Scientific). Each 1 mg/ml of full length FilP equals 29 $\mu$M.

#### *Affinity purification of N- and C-terminally 6×His-tagged FilP*

All of the bacterial strains containing tagged FilP were grown and harvested in the same way as described for the non-tagged FilP strain. Cells were thawed on ice, resuspended with buffer I (0.01 M sodium phosphate buffer, 0.01 M Tris–HCl, pH 8, and 8 M urea) and incubated for 1 h at RT. Lysed cells were centrifuged at 10,000$g$ for 30 min. The supernatant was incubated with TALON metal affinity resin (BD Biosciences) for 30 min at RT. TALON resins were washed with buffer II (0.01 M sodium phosphate buffer, 0.01 M Tris–HCl, pH 6.8, and 8 M urea). FilP was eluted from TALON resin with buffer IV (0.01 M sodium phosphate buffer, 0.01 M Tris–HCl, pH 5.9, and 8 M urea) followed by buffer V (0.01 M sodium phosphate buffer, 0.01 M Tris–HCl, pH 4.5, and 8 M urea).

### Dialysis buffers

For each dialysis assay, denatured FilP in 6 or 8 M urea was dialyzed against the favored buffer (see below). The dialysis was performed on 0.025 $\mu$m VSWP membrane filters (Millipore) at 4°C for 1 h in 20-ml plastic Petri dishes containing dialysis buffer.

#### *FilP solubility and structure in buffers at different pH*

FilP solubility was assessed through dialysis in four different buffers: 20 mM citric acid (Sigma-Aldrich), pH 4; 50 mM Tris–HCl (Sigma-Aldrich), pH 6.8 and 8.8; and 20 mM ethanolamine (Sigma-Aldrich), pH 11.

### FilP solubility and structure in different buffering systems

Dialysis of FilP was performed against four different buffers: 50 mM Tris–HCl (Sigma-Aldrich), pH 6.8; 50 mM sodium phosphate (Sigma-Aldrich), pH 7; 20 mM Hepes (Sigma-Aldrich), pH 7; and polymix buffer, pH 7. The polymix buffer is composed of 5 mM magnesium acetate (Sigma-Aldrich), 0.5 mM calcium chloride (Sigma-Aldrich), 8 mM putrescine (Sigma-Aldrich), 1 mM spermidine (Sigma-Aldrich), 5 mM potassium phosphate (Sigma-Aldrich), 95 mM potassium chloride (Sigma-Aldrich), 5 mM ammonium chloride (Sigma-Aldrich), and 1 mM DTT (VWR).

### FilP solubility and structure in the presence of salts

For this assay, dialysis buffers with different salts were prepared; 50 mM Tris–HCl, pH 6.8, and the addition of one or a combination of the following salts: 0.2 M KCl, 5 mM $MgCl_2$, and/or 1 M NaCl.

### Negative staining EM

Samples (3.5 $\mu$l) were applied on carbon-coated and glow-discharged 300 mesh copper grids (SPI) and adsorbed for 1 min. The samples were then washed in two drops of 50 $\mu$l deionized water and stained with 50 $\mu$l of 1.5% uranyl acetate for 30 s. Negatively stained samples were visualized by TEM: JEOL 1230 operating at 80 kV and Philips CM120 or FEI Talos L120 operating at 120 kV. Digital images were recorded with Gatan Orios 4k-pixel CCD camera and Digital Micrograph software, Olympus CCD SIS Cantega G2 2k-pixel CCD camera using iTEM software, or FEI Ceta 16k-pixel CMOS camera and TIA software. Image brightness and contrast were adjusted using Adobe Photoshop CC software.

### E. coli fixation and sectioning for EM

Induced *E. coli* cells containing recombinant FilP in pET28a plasmid were harvested and fixed with 2.5% glutaraldehyde (TAAB Laboratories Equipment) and 4% paraformaldehyde (Thermo Fisher Scientific), embedded in LR white resin (Sigma-Aldrich), and processed with an ultramicrotome (Leica EM UC7) into 70-nm thin sections. Sections were contrasted using uranyl acetate (Polysciences) and lead citrate (Thermo Fisher Scientific). Images were collected with Talos L120 TEM (FEI) using Ceta CMOS detector (FEI).

### Turbidity measurements

Samples were dialyzed against the desired buffer in Slide-A-Lyzer mini dialysis units with 10,000 MWCO (Thermo Fisher Scientific) overnight at 4°C. Dialyzed samples were transferred to 190 $\mu$m clear-bottomed, 96-well microplates (Greiner Bio-One). Turbidity, as a value for total light scattering caused by filamentous polymers, was measured at a 350 nm wavelength with TECAN infinite M200. Unassembled FilP in 6 M urea was used as a control in all turbidity experiments because FilP did not form filaments in the presence of 6 M urea and the absorbance of completely denatured FilP at 350 nm was zero.

### In vivo concentration estimation

Wild-type *S. coelicolor* culture (60 ml) was harvested and washed three times with 1 × PBS buffer, resulting in 0.3 g of washed bacteria (wet weight). The pelleted bacteria were resuspended with lysis buffer (0.1 M phosphate buffer, pH 7, 0.15 M NaCl), sonicated, and disrupted at 4°C. Known concentrations of recombinantly purified FilP (7.8–62.5 ng) were used as standards. Standard samples alongside washed *S. coelicolor* mycelia were separated on 10% SDS–PAGE. Proteins were transferred to PVDF membranes (Bio-Rad) based on the manufacturer's protocol. Polyclonal rabbit anti-FilP (Söderholm et al, 2018) was used as a primary antibody, and horseradish peroxidase-conjugated antirabbit antibodies were used for detection. Blots were visualized using ECL reagents and scanned with a C-DiGiT blot scanner (LI-COR). Band intensities were measured via IMAGE STUDIO software. The measured bacteria sample band intensity was used to calculate a standard line for each experiment and a value for FilP mass. The result was averaged from four individual experiments.

### Protein sedimentation assay

His-N-FilP was diluted to 200 $\mu$g/ml. Diluted samples were dialyzed against the favored buffer. The dialyzed samples (40 $\mu$l) were centrifuged at 135,000–300,000$g$ for 15–30 min at 25°C (Beckman Optima Max-XP, TLA–100 rotor). The supernatant of each sample was separated from the pellet, and the pellets were solubilized with 40 $\mu$l of the same dialysis buffer for 15 min at RT. Negative staining was done on both supernatants and pellets. The remaining supernatant was used for protein concentration quantification.

### Statistical analysis

Statistical analyses for turbidity and sedimentation assays are based on the average of three experiments and individual measurements for each point on the graph. Error bars represent the SD, and *P*-values were calculated using the one-way ANOVA test.

### Single-particle classification

FilP proteins were diluted in ethanolamine, pH 11, to a final concentration of 200 $\mu$g/ml and separated via size-exclusion chromatography (Superdex 75, 10/300 GL; GE Healthcare). A single fraction of the peak with homogenous FilP was collected from the column elution, negatively stained, and imaged by Talos L120 TEM (FEI) with Ceta CMOS detector (FEI). Particle picking and data processing were performed with the Scipion software package (de la Rosa-Treviń et al, 2016). Datasets with 388 extracted particle images (p.i.) of FilP were processed grouped into four classes using the Relion (2D classification) algorithm (Scheres, 2012a, 2012b; Scheres & Chen, 2012). After exclusion of 72 p.i. in the nonrepresentative class, further 2D classification of the 316 p.i. into three classes was performed. The length of particles in the representative classes were measured using ImageJ software (Schneider et al, 2012).

### Cryo-EM and tomography

In standard assembly buffer, freshly dialyzed 200 $\mu$g/ml of FilP filaments were mixed with 1.8 nm Ni-NTA-Nanogold (Nanoprobes) or 10 nm gold fiducials for tomography. A drop of 4 $\mu$l of the sample

was applied to 2/1 or 2/2 holy carbon film on 200 mesh copper grids (Quantifol) and vitrified by plunge freezing in liquid ethane using the Vitrobot system (FEI). Grids were transferred to the autoloader cassette to be analyzed with Titan Krios 300 kV cryo-TEM (FEI). Images were recorded with the K2 BioQuantum direct electron detector (Gatan). Nano-goldlabeled FilP images were collected at 130,000× magnification with a dose of 20 e$^-$/Å$^2$. The model of nano-gold localization was created with Photoshop. Tilt series were collected at 35,000× magnification with 2° increment automatically with the Tomography software (FEI) over an angular range of –60° to +60°, with a total dose of ~80 e$^-$/Å$^2$. Frames were motion-corrected with MotionCorr2. 3D reconstruction, modeling, and visualization were completed with the IMOD software suite (Kremer et al, 1996).

## Supplementary Information

## Acknowledgements

This study was performed in memory of Nora Ausmees, who contributed to inspiration, initial ideas, and design of the project. For technical support, we acknowledge the EMBL-Heidelberg EM facility for microscopy access, Umeå Core Facility for Electron Microscopy, the SciLifeLab national cryo-EM at Umeå University, National Microscopy Infrastructure (NMI, Ref no 2016-00968) for EM access, including instruments funded by the Knut and Alice Wallenberg foundation and Kempe foundations. Arash Javadi is acknowledged for graphical drawings. We thank the EMBL-Heidelberg Protein Purification and Expression Facility and Günter Stier for pETM plasmids. This project was funded by the Swedish Research Council project grant to L Sandblad (Ref no 2011-05198) Umeå Centre for Microbial Research (UCMR, Ref no 349-2007-8673), The Laboratory for Molecular Infection Medicine Sweden (MIMS, Ref no 2016-06598), and Gunnar Öquist fellowship from the Kempe foundations.

### Author Contributions

A Javadi: conceptualization, data curation, formal analysis, validation, investigation, visualization, methodology, and writing—original draft, review, and editing.
N Söderholm: conceptualization, data curation, formal analysis, validation, investigation, methodology, and writing—original draft, review, and editing.
A Olofsson: data curation and formal analysis.
K Flärdh: conceptualization and writing—original draft, review, and editing.
L Sandblad: conceptualization, data curation, formal analysis, supervision, funding acquisition, validation, investigation, visualization, methodology, project administration, and writing—original draft, review, and editing.

### Conflict of Interest Statement

The authors declare that they have no conflict of interest.

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
