## [Reviewer comments · Life Science Alliance]

Life Science Alliance

Assembly mechanisms of the bacterial cytoskeletal protein FilP

Ala Javadi, Niklas Söderholm, Annelie Olofsson, Klas Flärdh, and Linda Sandblad

DOI: <https://doi.org/10.26508/lsa.201800290>

Corresponding author(s): Linda Sandblad, Umeå University

Review Timeline:

Submission Date:	2018-12-28
Editorial Decision:	2019-02-18
Revision Received:	2019-05-26
Editorial Decision:	2019-06-05
Revision Received:	2019-06-13
Accepted:	2019-06-14

Scientific Editor: Andrea Leibfried

Transaction Report:

February 18, 2019

Re: Life Science Alliance manuscript #LSA-2018-00290

Dr. Linda Sandblad
Umeå University
Department of Molecular Biology
Försörjningsvägen
Umeå 901 87
Sweden

Dear Dr. Sandblad,

Thank you for submitting your manuscript entitled "Assembly mechanisms and structural conformations of the bacterial cytoskeletal protein FilP" to Life Science Alliance. The manuscript was assessed by expert reviewers, whose comments are appended to this letter. Please excuse the delay in getting back to you. A third review was repeatedly promised for your work but not delivered. I therefore now decided to move forward without it.

As you will see, the reviewers appreciate your work but think that your conclusions are currently not sufficiently supported by the data provided. They provide constructive input on how to strengthen your work, and we would thus like to invite you to submit a revised version of your work, addressing the reviewers' concerns. Importantly, the reviewers point out that critical controls for the used constructs are missing (N-terminal His-tagged protein), and that direct experimental evidence must be provided for an a-polar assembly model (or such model omitted).

Thank you for this interesting contribution to Life Science Alliance. We are looking forward to

receiving your revised manuscript.

Sincerely,

B. MANUSCRIPT ORGANIZATION AND FORMATTING:

Reviewer #1 (Comments to the Authors (Required)):

Javadi et al. characterized the in vitro assembly of the FilP protein. They analyzed the assembly of the N-terminus His-tagged protein since they suggested that it assembled as the WT protein. Using negatively stained FilP assemblies the authors studied the paracrystalline fibres of this filament. These assemblies are similar to lamins, which tend to assemble into paracrystals fibers in vitro, and when over expressed in cells. Different assembly buffers resulted in different macroscopic assembly nature. This basic characteristic information on the protein is very interesting. However, I find some inaccuracies as well as conclusions that may not exclusively reflect the results:

1. The title of this manuscript suggests that the authors can identify the conformations of the FilP protein. The resolution of the study presented here, is too low and only the assembly of filaments are shown (not the conformations of the protein).
2. Intermediate filament proteins are classified into 6 classes and not 5 as stated. Class 6 is nestin.
3. "...forming a 10-17 nm thick filament". It is well established that IF filaments are assembled into ~10 nm filaments. Averaged measurements never showed 17 nm. In Stromer et al. 1987 a mean up to 15nm was measured !
4. The authors compared the assembly of His-N, C-His and no tagged protein. At 20ug/ml the His-N assembled (60 nm repeats) very different than the C-His and untagged protein (~30nm repeats). This observation suggests that using the a his-tag of the N-terminus of the protein is problematic. It is not surprising because the N-terminus is very important also for lamin assembly. Moreover, the physiological concentrations of the protein was calculated to be 63ug/ml, therefore it is not clear how the authors concluded that His-N-FilP assembled as the WT protein.
5. Plastic section TEM images of E. Coli, over-expressed the C-His and no tagged FilP would be very important to compare and show the relevance of the N-His assembly, if at all.
6. The figure shown the FilP (no tag) assembly at 200ug/ml is at very low quality.
7. The authors try to make an analogy between FilP assembly and cytoplasmic filament assembly (e.g., vimentin). However, from the papcrystal assembly (for example 4C) it is clear that the filament assembled very similar to lamins. As in lamin paracrystals, individual protofilaments can be traced for long distances. Similarly, the protofilament in lamin paracrystals are not well organized and interact with each others (See cryo-ET lamin paracrystals, Taimen et al, 2009, Ben Harussh 2009, Turgay et al .2017 Supp). Therefore, all the results support the notion that FilP assembled into head-to-tail filaments. Indeed, no cytoplasmic IFs tend to form paracrystalline fibers.
8. The authors suggested that the diameter of the building blocks of the filaments are ~5nm. This would mean that the structure is hexameric in cross-section (Zaccai et al., 2011). This is very different than lamin as well as other IF proteins.
9. The authors discuss lamin filaments and protofilaments. Here they also cite Mahamid et al. However, Mahamid et al suggested that lamins are "dimers of coiled-coils assembled head-to-tail" (namely polar) with an average length of 67nm (Table S3)- if the structure resolved there was lamins they are not filamentous. Therefore, does not represent lamin assembly not the assembly of FilP.
10. The model suggests that the building blocks are interacts in an a-polar manner. Why is that ? there are no evidence for this assumption. Lamin become a-polar by interaction of polar filaments directed at opposite directions

Reviewer #2 (Comments to the Authors (Required)):

Intermediate filaments (IF) and IF-like proteins form a major class of cytoskeletal proteins that serve diverse roles. Long known functions include, among others, stabilization of the nuclear envelope in eukaryotes (see lamins), while more recently discovered members of the family have roles in cell

shape determination in bacteria (see crescentin). Javadi et al provide an in vitro characterization of the assembly of the IF-like protein FilP of *Streptomyces coelicolor*, which is involved in growth and morphogenesis in this bacterium. The authors primarily utilize FilP that was tagged at its amino terminal with a hexahistidine tag and was isolated from *E. coli*, but also perform experiments with untagged FilP, with C-terminal tagged FilP, and FilP isolated from *S. coelicolor*. They characterize the assembly properties of the protein and their dependence on protein concentration, level of denaturation (urea concentration), pH, and metal cation presence. They utilize a series of EM techniques, including cryo-electron tomography, to characterize the properties of the assembled protein at high resolution. These assembled forms include striated filament bundles with a repeat unit of about 60 nm, small rods with a length of 37 nm, and hexagonal sheets formed primarily in the presence of salts. Based on their findings, the authors propose an assembly model that includes assembly of the 37 nm units into protofilaments and their lateral association into larger structures. This work is comprehensive and technically sound. The study is thus informative to the understanding of bacterial IF-like proteins and their in vitro assembly properties. Indeed, the model the authors propose is novel in the case of bacterial IF-like proteins. The strength of the findings, the experimental support for the model, and the interest the current paper could generate could be increased in a few ways (see major comments), and the paper could be further improved by addressing some less important points (see minor comments).

Major comments

- 1) Can the assembly of His-N-FilP into striated filaments reflect the assembly of FilP in wild type *S. coelicolor* cells? Can His-N-FilP and FilP-C-His complement the defects of the Δ filP strain? Their localization (Fig. 1A) should be shown at higher magnification / better resolution. Is the grayer aspect of the cells in Fig 1B due to different focus or is it a phase contrast difference due to incomplete functionality of FilP-C-His?
- 2) The authors propose a model in which the amino termini of FilP form the major bands observed in the EM images, while the carboxyl termini form the minor bands. The reasoning the authors provide based on changes in relative band prominence between His-N-FilP and FilP-C-His filaments, while logic, is insufficient. More direct evidence that this is the orientation of the units would be more than welcome and would make the model convincing. Can the His tags be used to specifically indicate their position on the filament, either using anti-His antibodies fused to gold particles, or using metal based labeling, or any other technique that would directly answer this question?
- 3) The filament bundles described in the text are significantly different from the hexagonal structures primarily obtained in the presence of salts. Which ones are found in *S. coelicolor* cells? Can cryo-electron tomography of *S. coelicolor* reveal native structures? It would be of great interest to understand which structures are relevant in vivo, although this question may also present technical challenges. At minimum, a more detailed discussion of the in vivo relevance of the in vitro FilP assemblies is warranted.
- 4) Were the "native" His-N-FilP filaments (Fig 1D) obtained by purification of monomers from the cell lysate supernatants, which then assembled into filaments, or is there experimental evidence that they are filaments that existed inside the cells and were purified as such?

Minor comments

- 5) Please include page and line numbers. Reviewing the paper is quite difficult in their absence
- 6) While the English is generally well written, there are quite a few errors throughout the text. A thorough grammar and language check would help.
- 7) There are obvious differences between the filaments formed by FilP at 20 and 200 μ g/mL (Fig. 2C), as well as between those formed by FilP and those formed by His-N-FilP and FilP-C-His (Fig.

- 2A-C). These differences should be discussed more extensively alongside the support lent to the proposed assembly model by the similarities seen in the assembly patterns.
- 8) It would be very informative to readers if the authors summarized in a table the various measurements of the repetitive units in the various conditions tested. They could also include published representative measurements of the same distances obtained using other IFs.
 - 9) The authors should provide details on how bacterial strains were grown, how bacterial strains were generated, and how pellets for protein extraction were obtained. It almost looks like a full page from the methods section, between cloning and protein purification, is missing.
 - 10) I'm not sure all abbreviations are defined in the text, as they should be.
 - 11) Protein concentrations are given in $\mu\text{g}/\text{mL}$. What are the equivalent molar concentrations and how do they compare with concentrations used in similar assays on other IFs and IF-like proteins?
 - 12) All figure legends: what do P values represent and how were they calculated. There should be a statistics section among the other methods.
 - 13) Figure 6A: It appears that the images are circular cutouts placed on a gray background. The separation between the actual image and the background should be made obvious, lest the background be erroneously interpreted as being part of the original image.
 - 14) Figure S1A: can a larger part of the gel be shown to better support the claim of high purity of the preparation?
 - 15) Is Figure S2C called anywhere in the text?
 - 16) How do the His-N-FilP and FilP-C-His constructs used here differ from His-tagged FilP described in previous publications?

Reviewer #1 (Comments to the Authors (Required)):

Javadi et al. characterized the *in vitro* assembly of the FilP protein. They analyzed the assembly of the N-terminus His-tagged protein since they suggested that it assembled as the WT protein. Using negatively stained FilP assemblies the authors studied the paracrystalline fibres of this filament. These assemblies are similar to lamins, which tend to assemble into paracrystals fibers *in vitro*, and when over expressed in cells. Different assembly buffers resulted in different macroscopic assembly nature. This basic characteristic information on the protein is very interesting. However, I find some inaccuracies as well as conclusions that may not exclusively reflect the results:

1. The title of this manuscript suggests that the authors can identify the conformations of the FilP protein. The resolution of the study presented here, is too low and only the assembly of filaments are shown (not the conformations of the protein).

Response: The phrasing "and structural conformations" was removed from the title. Text page 1 line 1-2.

2. Intermediate filament proteins are classified into 6 classes and not 5 as stated. Class 6 is nestin.

Response: The Sixth class of IF, nestin, was added to the introduction. Text page 2 line 42-45.

3. "...forming a 10-17 nm thick filament". It is well established that IF filaments are assembled into ~10 nm filaments. Averaged measurements never showed 17 nm. In Stromer et al. 1987 a mean up to 15nm was measured !

Response: Text was changed to "approximately 10 nm thick filaments" in the introduction page 2 line 58.

4. The authors compared the assembly of His-N, C-His and no tagged protein. At 20µg/ml the His-N assembled (60 nm repeats) very different than the C-His and untagged protein (~30nm repeats). This observation suggests that using the a his-tag of the N-terminus of the protein is problematic. It is not surprising because the N-terminus is very important also for lamin assembly. Moreover, the physiological concentrations of the protein was calculated to be 63µg/ml, therefore it is not clear how the authors concluded that His-N-FilP assembled as the WT protein.

Response: To address the problems with protein structure studies with tagged proteins, we investigated the effect of cellular localization, function and *in vitro* polymerization of a filamentous protein with a N- or C-terminal His-tag, as well as the non-tagged protein. We show how these different constructs assemble and highlight the variation in structure, which we hope the readers appreciate to compare.

In contrast to lamin, an N-terminal His-tag stabilizes FilP filaments. We have added a new FilP construct; truncated at the N-terminal lacking the head and first coiled-coil domain (FilP aa 71-310). It still forms filament bundles with the same repetitive striation unit, but the major bands are protein dense. So for FilP, the N-terminus is not as important for assembly as it is for lamin assembly. Filaments formed by the 71-310 construct was added to this manuscript in Supplementary figure S2C. Text page 6 line 170-179. However, the C-terminus is more important for filament formation. With a C-terminally tag the filament structure varies at high and low concentrations. When removing the C-terminal tail and end coiled-coil domain, filamentation fails (not presented here, since this is part of a future manuscript).

Yes, for the different constructs in Figure 2 the repetitive striation pattern differs, but have common factors: The approximately 60 nm repeat for N-His-FilP and approximately 30 nm repeat FilP-C-His, which fit to the same pattern; two times the 30 nm repeat fits within the 60 nm striation pattern. This has been revised in the text explained with an additional Supplementary figure S2C. Text page 6 line 170.

We have looked at filament assembly at different concentrations for all constructs. Protein filament bundle/paracrystal structure is identical for all concentrations tested for N-His-FilP. In the manuscript we present images and turbidity studies for *in vitro* concentrations close; slightly lower 20 µg/ml and slightly higher 200 µg/ml, than the physiological concentration 63 µg/ml. Since cellular concentrations of FilP is concentrated at the hyphal tips, investigation of higher and lower concentration than the average cellular concentration is relevant and useful for the reader and future studies.

5. Plastic section TEM images of *E. Coli*, over-expressed the C-His and no tagged FilP would be very important to compare and show the relevance of the N-His assembly, if at all.

Response: Plastic sections of *E. coli* overexpressing recombinant FilP-C-His and non-tagged FilP have been added to the manuscript, images are in Supplementary figure S2D and E. Text page 6 line 190-192.

6. The figure shown the FilP (no tag) assembly at 200ug/ml is at very low quality.

Response: The protein form thick filament bundles at high concentrations, TEM signal has a low contrast when the sample is thick. The experiment and image analyze was repeated, a less thick filaments, which could be imaged with a more favorable contrast, still showing the same repetitive structure is used. A new image was inserted in Figure 2C.

7. The authors try to make an analogy between FilP assembly and cytoplasmic filament assembly (e.g., vimentin). However, from the papcrystal assembly (for example 4C) it is clear that the filament assembled very similar to lamins. As in lamin paracrystals, individual protofilaments can be traced for long distances. Similarly, the protofilament in lamin paracrystals are not well organized and interact with each others (See cryo-ET lamin paracrystals, Taimen et al, 2009, Ben Harussh 2009, Turgay et al .2017 Supp). Therefore, all the results support the notion that FilP assembled into head-to-tail filaments. Indeed, no cytoplasmic IFs tend to form paracrystalline fibers.

Response: According to the model of lamin "head-to-tail" assembly" each dimer has a predicted length similar to the repetitive unit of the paracrystalline bundle (24 nm: Taimen et al, 2009 / 48 nm, comprised of alternating staggered protofilaments, with a lamin dimer length of 55 nm: Ben Harussh 2009 / 20 nm: Turgay et al .2017 Supp.) Lamin also has an Ig-like tail domain, contributing to the protein density in the striation pattern, FilP has head- and tail-peptide sequences without a predicted structured fold. FilP has a predicted dimer length of 40 nm, in this manuscript called basic building block, we interpret this basic building block to be the rod-shaped molecule we observed at high pH. However, since this rod and predicted dimer is shorter than the 60 nm repetitive unit we observed, and since this repetitive bundle/paracrystal unit is symmetrical/non-polar, we conclude that FilP does not assemble head-to-tail as lamin, instead a tail-to-tail and head-to-head assembly would match the filament confirmation. So FilP display a paracrystalin bundle structure similar to lamin. And FilP is predicted to have a primary assembly arrangement similar to what was predicted for vimentin and desmin. We have rewritten this in the text to be clearer about how we understand/interpret the assembly results. Text page 11 line 351, page 12 line 391-393, page 13 line 413-414, page 14 line 456-461.

8. The authors suggested that the diameter of the building blocks of the filaments are ~5nm. This would mean that the structure is heaxmeric in cross-section (Zaccai et al., 2011). This is very different than lamin as well as other IF proteins.

Response: The negative staining EM is not the best method to measure the thickness of a filaments, since during preparation the stain is deposit around the proteins, creating a negative contrast. In relation to other findings this measurement is less important and we chose to remove it.

9. The authors discuss lamin filaments and protofilaments. Here they also cite Mahamid et al. However, Mahamid et al suggested that lamins are "dimers of coiled-coils assembled head-to-tail " (namely polar) with an average length of 67nm (Table S3)- if the structure resolved there was lamins they are not filamentous. Therefore, does not represent lamin assembly not the assembly of FilP.

Response: Mahamid et al. visualize 67 nm lamin (with a variation of ± 39 , and assumed to be polar) molecules by cellular electron tomography, we appreciate that the article gives this view of the physiological structure. Molecules, which according to a lamin coiled-coil protein length could be both dimers and longitudinally interacting polymers. We refer to Mahamid 2016 (together with Abei 1986 and Turgay 2017) which are studies visualizing lamin in their natural cells, because they all report lamin structure in form of thin filaments in less organized meshwork. In the cellular volumes referred to, no thick bundles and paracrystals were visible. However, also no hexagonal meshwork, as FilP has the ability to assemble into, has been reported for lamin, which is another difference. Lamin dimers has the length to bridge transversal between two striation bands in a paracrystal, which explains that "dimers of coiled-coils assembled head-to-tail". However, FilP is shorter, approximately 40 nm, and would theoretically require 2 x coiled-coil dimer to bridge between the striation bands with a 60 nm repeating unit. This means that 2 x dimer should be non-polar arranged within a repetitive unit. In our images, reparative units appear to be non-polar/symmetrical, also filaments appear to have two ends with similar structure. We hope this explains the difference between lamin and FilP. To clarify, we have changed the wording in the discussion page 12 line 391-393 and page 14 line 456-461.

10. The model suggests that the building blocks are interacts in an a-polar manner. Why is that ? there are no evidence for this assumption. Lamin become a-polar by interaction of polar filaments directed at opposite directions

Response: Since the protofilament/bundle repetitive unit is longer than the basic building block, see reply to comment 4 and 9, there is no sign of FilP polar protofilament letteral assembly in opposite directions in our negative staining images and cryo electron tomogram, also nickel-gold labeling on FilP bundles are symmetrical according to the paracrystal repetitive unit. We predict that the protofilament repetitive unit is non-polar, build up

by tail-to-tail interactions of two basic building blocks/presumably coiled-coil dimers. Text page 11 line 351-361, page 14 line 456-461.

Reviewer #2 (Comments to the Authors (Required)):

Intermediate filaments (IF) and IF-like proteins form a major class of cytoskeletal proteins that serve diverse roles. Long known functions include, among others, stabilization of the nuclear envelope in eukaryotes (see lamins), while more recently discovered members of the family have roles in cell shape determination in bacteria (see crescentin). Javadi et al provide an in vitro characterization of the assembly of the IF-like protein FilP of *Streptomyces coelicolor*, which is involved in growth and morphogenesis in this bacterium. The authors primarily utilize FilP that was tagged at its amino terminal with a hexahistidine tag and was isolated from *E. coli*, but also perform experiments with untagged FilP, with C-terminal tagged FilP, and FilP isolated from *S. coelicolor*. They characterize the assembly properties of the protein and their dependence on protein concentration, level of denaturation (urea concentration), pH, and metal cation presence. They utilize a series of EM techniques, including cryo-electron tomography, to characterize the properties of the assembled protein at high resolution. These assembled forms include striated filament bundles with a repeat unit of about 60 nm, small rods with a length of 37 nm, and hexagonal sheets formed primarily in the presence of salts. Based on their findings, the authors propose an assembly model that includes assembly of the 37 nm units into protofilaments and their lateral association into larger structures.

This work is comprehensive and technically sound. The study is thus informative to the understanding of bacterial IF-like proteins and their in vitro assembly properties. Indeed, the model the authors propose is novel in the case of bacterial IF-like proteins. The strength of the findings, the experimental support for the model, and the interest the current paper could generate could be increased in a few ways (see major comments), and the paper could be further improved by addressing some less important points (see minor comments).

Major comments

1) Can the assembly of His-N-FilP into striated filaments reflect the assembly of FilP in wild type *S. coelicolor* cells? Can His-N-FilP and FilP-C-His complement the defects of the Δ filP strain? Their localization (Fig. 1A) should be shown at higher magnification / better resolution. Is the grayer aspect of the cells in Fig 1B due to different focus or is it a phase contrast difference due to incomplete functionality of FilP-C-His?

Response: Yes, all experiments indicate that His-N-FilP function as the wild type protein in *S. coelicolor*. And since FilP has a strong tendency to polymerize, even at low concentrations, we believe it is likely that FilP form filaments or meshwork in *Streptomyces* bacteria. Filament bundles with high periodicity is easy to detect by EM but meshwork have lower contrast and may be hard to detect in thin EM sections.

According to the reviewer #2 suggestion a complementation experiment was performed. We were able to show complementation in the Δ filP strain by His-N-FilP. No phenotype complementation occurred when the 6xHis-tag is fused to the c-terminal of the protein. However, the localization of FilP-C-His is identical to the wild type (WT) (Figure 1B). Images of the complementation experiment were added to Supplementary figure S1D-E. Text page 4 line 124-128. After addition of this new Supplementary figure S1, all following supplementary material is shifted one digit.

We chose to insert magnified views of *S. coelicolor* hypha tips in the Supplementary figure 1A-C. Text page 4 line 124.

The low contrast in Figure 1B giving the phase contrast (Ph) image a "grey" appearance, is not caused by the bacteria culture, growth or protein content. All cultures in the experiment look similar by ocular inspection. The low contrast is caused by the auto contrast function of the microscope/camera system. Since the contrast does not affect the result we choose not to adjust the contrast. If required for publishing, the contrast could be adjusted by photoshop, without interfering with the view and results, please let us know if we should do so.

2) The authors propose a model in which the amino termini of FilP form the major bands observed in the EM images, while the carboxyl termini form the minor bands. The reasoning the authors provide based on changes in relative band prominence between His-N-FilP and FilP-C-His filaments, while logic, is insufficient. More direct evidence that this is the orientation of the units would be more than welcome and would make the model convincing. Can the His tags be used to specifically indicate their position on the filament, either using anti-His antibodies fused to gold particles, or using metal based labeling, or any other technique that would directly answer this question?

Response: As Reviewer #2 suggested, we have used 1.8 nm nickel NTA Nanogold to label the 6xHis tags in the filament bundles, images of FilP with gold were collected by cryo-EM. This experiment gives a clear and undoubtable view of nickel-6xHis affinity interaction to the major band in N-His-FilP bundles. FilP-C-His filament bundles are more disordered with labeling spread out over the filaments, which could indicate that the C-terminal is flexibly structured or that Nanogold binding to FilP could be unspecific. Since Nanogold displays a faint affinity to bundles of non-tagged FilP, we speculate that unspecific localizations of gold to FilP (not nickel-6xHis dependent) make the images unclear for both for FilP-C-His and non-tagged protein. Figures were added to Supplementary

Figure S2D-E. Text page 11 line 353-361, page 14 line 432-439 and 444-449, page 22 line 718-719 and 723-725, and Supplementary figure S3 legend page 27 line 877-887.

3) The filament bundles described in the text are significantly different from the hexagonal structures primarily obtained in the presence of salts. Which ones are found in *S. coelicolor* cells? Can cryo-electron tomography of *S. coelicolor* reveal native structures? It would be of great interest to understand which structures are relevant in vivo, although this question may also present technical challenges. At minimum, a more detailed discussion of the in vivo relevance of the in vitro FilP assemblies is warranted.

Response: To visualize FilP structure in living bacteria is a high priority aim for our lab, but as reviewer #2 stated it is technically challenging, we have worked with high pressure freezing and lowacryl embedding for immunolabeling of FilP without results for a clear FilP identification and localization. Also, cryo-electron tomography methods are a very good idea, we are working with that, currently we have not managed to get the acquired resolution to visualize FilP structures inside *S. coelicolor*. It is a challenge we hope to address in future papers. Further discussion to FilP assembly in a cellular context was added to the discussion. Text page 15 line 597-598, page 16-17 line 529-537.

4) Were the "native" His-N-FilP filaments (Fig 1D) obtained by purification of monomers from the cell lysate supernatants, which then assembled into filaments, or is there experimental evidence that they are filaments that existed inside the cells and were purified as such?

Response: For Figure 1D *Streptomyces* expressing N-His FilP were lysed without addition of urea in the buffers (which we normally use for FilP purification from *E. coli*). The cleared (membranes removed by centrifugation) lysate was incubated with TALON (cobolt) resin and eluted with Imidazol. The eluted fraction was dialyzed to remove imidazole and without urea denaturation bound to EM grids. With "native" we mean that the protein was never denatured during lysis and EM, so it is subsequently possible to assume that the protein fold and conformation could be the same in the EM grid as it may exhibit in the bacteria before "native" purification. Text updated for clarification page 5 line 145-147 and figure legend page 23 line 761-762.

Minor comments

5) Please include page and line numbers. Reviewing the paper is quite difficult in their absence

Response: Page and text line number were inserted in the manuscript.

6) While the English is generally well written, there are quite a few errors throughout the text. A thorough grammar and language check would help.

Response: Professional language correction have been used, and should be much better now.

7) There are obvious differences between the filaments formed by FilP at 20 and 200 $\mu\text{g}/\text{mL}$ (Fig. 2C), as well as between those formed by FilP and those formed by His-N-FilP and FilP-C-His (Fig. 2A-C). These differences should be discussed more extensively alongside the support lent to the proposed assembly model by the similarities seen in the assembly patterns.

Response: A comparative figure was added: Supplementary figure S2C. Text page 6 line 173-179, page 14 line 432-439.

8) It would be very informative to readers if the authors summarized in a table the various measurements of the repetitive units in the various conditions tested. They could also include published representative measurements of the same distances obtained using other IFs.

Response: A table of the repetitive unit length was added along the representative images in Supplementary figure S2C. Text page 6 line 170. A table comparing other IFs repetitive length was not added.

9) The authors should provide details on how bacterial strains were grown, how bacterial strains were generated, and how pellets for protein extraction were obtained. It almost looks like a full page from the methods section, between cloning and protein purification, is missing.

Response: Additional protocol information on bacteria strains and culturing and protein expression conditions have been added to the Methods and material section. Text page 17 line 553-555, 560-567, page 18 line 587-589.

10) I'm not sure all abbreviations are defined in the text, as they should be.

Response: Abbreviations have been written in full length first time used in the text. 3D, His, SDS-PAGE, EM, SEM, LB, LA, TSB and TSA have been clarified.

11) Protein concentrations are given in $\mu\text{g}/\text{mL}$. What are the equivalent molar concentrations and how do they compare with concentrations used in similar assays on other IFs and IF-like proteins?

Response: Each 1 mg/ml of full length FilP equals 29 μM . The unit mg/ml or $\mu\text{g}/\text{ml}$ is used in our protocols; however, we added this explanation to the Methods and Material section. Text page 18 line 584-585.

12) All figure legends: what do P values represent and how were they calculated. There should be a statistics section among the other methods.

Response: An explanation to the statistical calculation was added to the Methods and material section. Page 21-22 line 697-701.

13) Figure 6A: It appears that the images are circular cutouts placed on a gray background. The separation between the actual image and the background should be made obvious, lest the background be erroneously interpreted as being part of the original image.

Response: During single particle 2D class averaging the software cut out each individual image, align the images in each class through 360° rotation, shown in Figure 6A are the average image of each class, circular cropping facilitates image rotation for 2D alignment. Figure 4D show a representative EM exposure with the molecules before averaging and the background of the entire exposed area. More EM raw data images, from the same experiment, which were included in the average images presented, could be attached in the supplementary material if the editor thinks it is needed.

14) Figure S1A: can a larger part of the gel be shown to better support the claim of high purity of the preparation?

Response: instead of a cropped image, Supplementary figure S2A was replaced with an image of the entire SDS-PAGE gel.

15) Is Figure S2C called anywhere in the text?

Response: The former Supplementary image of FilP in HEPES buffer was removed since it was not important for the results, and this modification makes the manuscript clearer, an image of FilP filaments in HEPES buffer is still presented in Figure 6B. Text page 9 line 286-290.

16) How do the His-N-FilP and FilP-C-His constructs used here differ from His-tagged FilP described in previous publications?

Response: Only the pETM28a-FilP plasmid for expression and purification of N-His-FilP was used in Bagchi et al, 2008 Figure 3A-B and in Fuchino et al, 2013 Figure 1. FilP-C-His and the non-tagged construct were never used in earlier publications, the origin of all plasmids and strains are presented in Supplementary table 4 (The table 4 was for this revision updated with the additional construct added to the manuscript FilP aa 71-310).

Thank You, best regards Linda Sandblad, on behalf of all authors

June 5, 2019

RE: Life Science Alliance Manuscript #LSA-2018-00290R

Dr. Linda Sandblad
Umeå University
Department of Molecular Biology
Försörjningsvägen
Umeå 901 87
Sweden

Dear Dr. Sandblad,

Thank you for submitting your revised manuscript entitled "Assembly mechanisms of the bacterial cytoskeletal protein FilP". As you will see, the reviewers appreciate the introduced changes but still raise some concerns that need your attention.

We would be happy to publish your paper in Life Science Alliance pending that you address the remaining reviewer concerns via text changes. Please also address the following editorial points:

- please list 10 authors et al in the reference list
- please add a description of panel S3F in the legend
- please rename the suppl table to Table S1 (currently S4) and upload as a docx file

A. FINAL FILES:

-- Summary blurb (enter in submission system): A short text summarizing in a single sentence the study (max. 200 characters including spaces). This text is used in conjunction with the titles of

papers, hence should be informative and complementary to the title. It should describe the context and significance of the findings for a general readership; it should be written in the present tense and refer to the work in the third person. Author names should not be mentioned.

B. MANUSCRIPT ORGANIZATION AND FORMATTING:

Sincerely,

Andrea Leibfried, PhD
Executive Editor
Life Science Alliance
Meyerohofstr. 1
69117 Heidelberg, Germany
t +49 6221 8891 502
e a.leibfried@life-science-alliance.org
www.life-science-alliance.org

Reviewer #1 (Comments to the Authors (Required)):

The revised version of Javadi et al. fulfilled most of my concerns, however, one correction should still be introduced :

1. "EM studies of cellular lamin configuration display a meshwork supporting the nuclear membrane, in the form of a woven network of protofilaments" . Again, Mahamid et al. does not fit here because no polymerized filament of lamin were detected- surely not protofilaments (defined as tetrameric structures at least as long as two building blocs- 100nm for lamins). However, if the authors (as they wrote in their rebuttal letter) would like to cite this paper as "give view of physiological structure" they should cite Herapin et al 2015 (Nature Methods), who were the 1st who reported on physiological view of lamins using FIB-SEM and cryo-ET.

Reviewer #2 (Comments to the Authors (Required)):

Javadi et al addressed most of my initial concerns to my satisfaction and improved their manuscript significantly. The complementation result, as well as the nanogold labeling strengthen the support for their model. The study generates a good quality analysis of the in vitro assembly of a bacterial IF-like protein, FilP, advancing the knowledge of this still understudied class of important proteins. Several issues remain, but they are likely easily addressed without the need for further experiments.

1. While the flow of the language and the presentation are good, some grammar and language mistakes remain. I did not catalog all of them individually. I suspect they can be easily corrected by the editorial staff at the proof stage.

2. Lines 106-110: please break up the convoluted sentence for better clarity.

3. The naming of the strains and of the constructs is not consistent. See lines 133 vs 146 vs Table S4 vs Fig 1 labels. Please correct.

4. Line 145: Use of the word "endogenously" is misleading: the constructs are not expressed from the endogenous locus, nor using the native promoter. Please rephrase.

5. Figure 1D: I appreciate the authors' clarification in their answer to my previous question (#4) that they used native (as in not denatured) protein that they obtained from *S. coelicolor* lysates.

However, another distinction is possibly more important. During their purification of His-N-FilP from *S. coelicolor*, the authors did not present any evidence that the filaments that they obtain at the end of the assay are purified as pre-formed filaments from the cell lysates, or are purified as monomers that then assemble into the shown filaments. The first scenario, in which a filament present in the lysate bound to the column and was then eluted, would support the claim that the filaments assembled in vitro look the same as the filaments assembled in the cells. The second scenario, in which monomers present in the lysate bind to the column, are then eluted, and only after elution assemble into filaments support the claim that natively folded protein assembles in vitro similarly to re-folded protein, but do not speak to whether the in vitro assembled filaments are the same as natively assembled filaments (i.e. filaments that assembled inside the *S. coelicolor* cells). This distinction is important and should be addressed, lest the readers become confused as to what the presented data actually shows.

6. The different contrast in Fig 1B compared to Fig 1A and 1C misleads the reader to see a phenotype where there isn't one. Autocontrast should not be used when comparing images from different samples. I assume the authors used the same acquisition parameters for the 3 samples (e.g. exposure time, power on the transillumination light source, etc.). If that is true, then the proper way for the images to be presented is to use the same lookup table for all 3 images used (of course, without saturating any part of any of the images), and not rely on the autoscale function of the

software. Please modify accordingly.

7. Lines 195-196: please rephrase such that the word filament is not used four times in one sentence

8. Line 276: I'm not sure the authors have shown that what they see in Fig 4D are molecules. I would recommend using a less definitive term to describe what they see in that sample.

9. Line 542: Since FilP is cytosolic, I would recommend the authors specify that any association with the peptidoglycan would be indirect. The current formulation seems to suggest that a direct interaction is invoked, which I do not believe was the authors' intention

10. Line 625: provide source of the antibody

11. Line 795: The authors have not made any filament flexibility measurements and therefore cannot claim that their data shows changes in flexibility. They do observe increased frequency of bends that could be caused by increased flexibility. Please rephrase to avoid overstating the results.

12. Line 869: Induced instead of incused

13: Lines 882 - 886. There seems to be some mistakes in the legend here. Please correct.

14. Figure 4A-D: Please clarify whether the different buffers differ in pH only, or do they also differ in ionic strength? If the later is true, does this affect in any way the interpretation of the results?

15. Figure 5B: It may be more clear to label the first column as "No addition" (to the buffer) instead of "buffer". The current labeling scheme may be read to mean that all the other columns were not done in buffer + the noted salts, but instead in the noted salts alone. Please adjust the figure legend accordingly as well.

16. I appreciate the author's clarification that Fig. 6A shows circular cutouts. Please include this information in the Figure legend as well. It is informative to the readers to know what they are looking at. Please also more obviously separate the circular cutouts from the fake gray background.

17. Figure S1E: The strain does not appear complemented, as the authors rightfully note in the text. But Fig. 1B looks complemented. Aren't these images of the same strain? Were different experimental conditions used? Please clarify.

18. Figure S3E,F: I cannot see any filament assembled. Can the authors play with the contrast of find another way to demonstrate that there really is a filament? Otherwise, the result's interpretation is not very solid. If there is significantly less protein bound to the grid, the nanogold may bind to the grid directly but non-specifically rather than specifically to the His tag on the protein, explaining the distribution of particles observed.

19. Table S4: use greek letter for alpha in DH5 α .

#Editor:

We would be happy to publish your paper in Life Science Alliance pending that you address the remaining reviewer concerns via text changes. Please also address the following editorial points:

- please list 10 authors et al in the reference list

Response: We have now used "EMBO" reference formatting

- please add a description of panel S3F in the legend

Response: We noticed S3F was in the manuscript accidentally wrong, the legend is now corrected indicated with (F)

- please rename the suppl table to Table S1 (currently S4) and upload as a docx file

Response: The table is now in .docx format and renamed to Table S1

Reviewer #1 (Comments to the Authors (Required)):

The revised version of Javadi et al. fulfilled most of my concerns, however, one correction should still be introduced :

1. "EM studies of cellular lamin configuration display a meshwork supporting the nuclear membrane, in the form of a woven network of protofilaments" . Again, Mahamid et al. does not fit here because no polymerized filament of lamin were detected- surely not protofilaments (defined as tetrameric structures at least as long as two building blocs- 100nm for lamins). However, if the authors (as they wrote in their rebuttal letter) would like to cite this paper as "give view of physiological structure" they should cite Herapin et al 2015 (Nature Methods), who were the 1st who reported on physiological view of lamins using FIB-SEM and cryo-ET.

Response: The sentence was rephrased to also include shorter polymers and reference was added according to suggestion by Reviewer #1. Page 13 text line 398-399.

Reviewer #2 (Comments to the Authors (Required)):

Javadi et al addressed most of my initial concerns to my satisfaction and improved their manuscript significantly. The complementation result, as well as the nanogold labeling strengthen the support for their model. The study generates a good quality analysis of the in vitro assembly of a bacterial IF-like protein, FilP, advancing the knowledge of this still understudied class of important proteins. Several issues remain, but they are likely easily addressed without the need for further experiments.

1. While the flow of the language and the presentation are good, some grammar and language mistakes remain. I did not catalog all of them individually. I suspect they can be easily corrected by the editorial staff at the proof stage.

Response: Yes, we are happy to make corrections.

2. Lines 106-110: please break up the convoluted sentence for better clarity.

Response: The sentences of the introduction was divided and reformulated. Page 4 text line 105-110.

3. The naming of the strains and of the constructs is not consistent. See lines 133 vs 146 vs Table S4 vs Fig 1 labels. Please correct.

Response: Is now corrected in page 4-5 line 132-133.

4. Line 145: Use of the word "endogenously" is misleading: the constructs are not expressed from the endogenous locus, nor using the native promoter. Please rephrase.

Response: "endogenous" was removed and the sentences was rephrased at page 5 lines 146-147.

5. Figure 1D: I appreciate the authors' clarification in their answer to my previous question (#4) that they used native (as in not denatured) protein that they obtained from *S. coelicolor* lysates. However, another distinction is possibly more important. During their purification of His-N-FilP from *S. coelicolor*, the authors did not present any evidence that the filaments that they obtain at the end of the assay are purified as pre-formed filaments from the cell lysates, or are purified as monomers that then assemble into the shown filaments. The first scenario, in which a filament present in the lysate bound to the

column and was then eluted, would support the claim that the filaments assembled *in vitro* look the same as the filaments assembled in the cells. The second scenario, in which monomers present in the lysate bind to the column, are then eluted, and only after elution assemble into filaments support the claim that natively folded protein assembles *in vitro* similarly to re-folded protein, but do not speak to whether the *in vitro* assembled filaments are the same as natively assembled filaments (i.e. filaments that assembled inside the *S. coelicolor* cells). This distinction is important and should be addressed, lest the readers become confused as to what the presented data actually shows.

Response: Yes, we intended to purify FilP filaments in their native folded form, folded in the cytoplasm of *S. coelicolor*. And clarified this in the text, page 4-5 line 132-133. To visualize purified FilP filaments under as native conditions as possible, we have not denatured or in any way disrupt the native conformation. The purification method from *Streptomyces* culture was further clarified in the methods section page 18 line 586-599.

However, it cannot be excluded that the filamentation properties may change during the affinity purification protocol. The minor changes of salt content, used during *in vitro* purification, in different buffer at neutral pH, does not (according to our results) affect dimerization, polymerization or the repetitive banding pattern.

The main reason to carry out this experiment, was to confirm that FilP expressed in *S. coelicolor* formed filaments equal to FilP expressed in *E. coli*. And that the structure of filaments, formed (after protein denaturation) by refolding, adopt the same structure as FilP folded in the bacteria cytoplasm and never denatured in urea.

6. The different contrast in Fig 1B compared to Fig 1A and 1C misleads the reader to see a phenotype where there isn't one. Autocontrast should not be used when comparing images from different samples. I assume the authors used the same acquisition parameters for the 3 samples (e.g. exposure time, power on the transillumination light source, etc.). If that is true, then the proper way for the images to be presented is to use the same lookup table for all 3 images used (of course, without saturating any part of any of the images), and not rely on the autoscale function of the software. Please modify accordingly.

Response: The contrast has now been adjusted so all 3 images 1A-C are displayed with comparable contrast. It is a good suggestion by reviewer #2 to present the figure so the reader can focus on the image and the result, not to confuse the reader with contrast variations. Image Figure 1B is updated and methods line 583.

7. Lines 195-196: please rephrase such that the word filament is not used four times in one sentence

Response: The sentence was rephrased, page 6 text line 194.

8. Line 276: I'm not sure the authors have shown that what they see in Fig 4D are molecules. I would recommend using a less definitive term to describe what they see in that sample.

Response: The word was replaced with "polymer units", page 9 line 275.

9. Line 542: Since FilP is cytosolic, I would recommend the authors specify that any association with the peptidoglycan would be indirect. The current formulation seems to suggest that a direct interaction is invoked, which I do not believe was the authors' intention

Response: Yes, FilP was earlier and in this study suggested to be localized in the cytoplasm.

However, it localizes to the sides of the hyphae tips and Walter et al 2003 suggest a membrane/peptidoglycan localization of a FilP-homologue in *S. lividans* and Söderholm et al 2018 demonstrated that carbohydrates bind to FilP. We change "peptidoglycan" to "carbohydrates" based on the suggestion from the reviewer, but prefer to keep the sentence to highlight the effects interaction with other molecules could have on the structure of FilP in the cell. Page 17 line 534

10. Line 625: provide source of the antibody

Response: The primary Ab is made by our lab (Söderholm et al 2018) and secondary Ab source was added to the methods section page, Page 18 line 572-574, page 22 line 699-700

11. Line 795: The authors have not made any filament flexibility measurements and therefore cannot claim that their data shows changes in flexibility. They do observe increased frequency of bends that could be caused by increased flexibility. Please rephrase to avoid overstating the results.

Response: The figure legend 2 was rephrased. Page 25 text line 798-799.

12. Line 869: Induced instead of incused

Response: Corrected in line 875.

13: Lines 882 - 886. There seems to be some mistakes in the legend here. Please correct.

Response: Supplementary figure legend 2 was corrected page 27 line 887-890

14. Figure 4A-D: Please clarify whether the different buffers differ in pH only, or do they also differ in ionic strength? If the later is true, does this affect in any way the interpretation of the results?

Response: The buffer conditions differ significantly in pH (from 3.9 to 11). There are minor differences in salt concentration, 20 to 50 mM and the Na⁺ and Cl⁻ ions added during buffer stabilization at the given pH (The buffer conditions are described in detail in Methods section text line 644-647), the variations in salt is relatively low (in the range of 10-50 mM). We know that variations around 10-50 mM Tris has no effect on the FilP filament structure. Larger differences in salt concentration is needed to affect the structure of FilP polymerization (0.2-1 M KCl or NaCl, see Figure 5). The same buffering system and exactly the same salt content was experimentally hard to accommodate.

15. Figure 5B: It may be more clear to label the first column as "No addition" (to the buffer) instead of "buffer". The current labeling scheme may be read to mean that all the other columns were not done in buffer + the noted salts, but instead in the noted salts alone. Please adjust the figure legend accordingly as well.

Response: Figure label and figure legend 5B was changed according to suggestion. Page 25 text line 826-829.

16. I appreciate the author's clarification that Fig. 6A shows circular cutouts. Please include this information in the Figure legend as well. It is informative to the readers to know what they are looking at. Please also more obviously separate the circular cutouts from the fake gray background.

Response: The images presented in Figure 6A are the outcome, created as a result of 2D classification by the software, we prefer to not modify them. An explanation was added to Figure legend 6. Text line 841-843.

17. Figure S1E: The strain does not appear complemented, as the authors rightfully note in the text. But Fig. 1B looks complemented. Aren't these images of the same strain? Were different experimental conditions used? Please clarify.

Response: It is the same strains and different experimental setups were used. Figure 1A-B are immunofluorescence images of *Streptomyces* grown on cellophane film on top of TSA to prevent bacteria from detaching during the staining procedure on glass and thereby obtain reproducible and reliable staining. The phenotype is not detectable when culturing the bacteria for immunofluorescence on cellophane. The complementation assay shown in S1D-E are performed through growth from the acute angle of a coverslip inserted into a TSA plate, a standard method used to differentiate the FilP deleted phenotype first described by Bagchi et al 2008.

The methods are now clarified in methods section under "light microscopy". Text page 17 line 562-565 and 575-681 and also the different culturing protocols are better clarified and described in the figure legends to Figure 1B-C and S1D-E.

18. Figure S3E,F: I cannot see any filament assembled. Can the authors play with the contrast of find another way to demonstrate that there really is a filament? Otherwise, the result's interpretation is not very solid. If there is significantly less protein bound to the grid, the nanogold may bind to the grid directly but non-specifically rather than specifically to the His tag on the protein, explaining the distribution of particles observed.

Response: An overview of holy carbon film holes was added for each sample, as a upper row, in figure S3D-F. This overview is recorded at a lover defocus giving the image a higher contrast so the amount of filament bundles are clearly visible, displaying equal amount of FilP filament bundles in all samples.

19. Table S4: use greek letter for alpha in DH5 α .

Response: OK, changed, in Table S1

June 14, 2019

RE: Life Science Alliance Manuscript #LSA-2018-00290RR

Dr. Linda Sandblad
Umeå University
Department of Molecular Biology
Försörjningsvägen
Umeå 901 87
Sweden

Dear Dr. Sandblad,

Thank you for submitting your Research Article entitled "Assembly mechanisms of the bacterial cytoskeletal protein FilP". I appreciate the introduced changes and it is a pleasure to let you know that your manuscript is now accepted for publication in Life Science Alliance. Congratulations on this interesting work.

DISTRIBUTION OF MATERIALS:

Again, congratulations on a very nice paper. I hope you found the review process to be constructive and are pleased with how the manuscript was handled editorially. We look forward to future exciting submissions from your lab.

Sincerely,
